# $f$-Divergence Variational Inference

**Neng Wan**[1][*]
nengwan2@illinois.edu

**Dapeng Li**[2][*]
dapeng.ustc@gmail.com

**Naira Hovakimyan**[1]
nhovakim@illinois.edu

[1] University of Illinois at Urbana-Champaign, Urbana, IL 61801
[2] Anker Innovations, Shenzhen, China

## Abstract

This paper introduces the $f$-*divergence variational inference* ($f$-VI) that generalizes variational inference to all $f$-divergences. Initiated from minimizing a crafty surrogate $f$-divergence that shares the statistical consistency with the $f$-divergence, the $f$-VI framework not only unifies a number of existing VI methods, *e.g.* Kullback–Leibler VI [1], Rényi's $\alpha$-VI [2], and $\chi$-VI [3], but offers a standardized toolkit for VI subject to arbitrary divergences from $f$-divergence family. A general $f$-variational bound is derived and provides a sandwich estimate of marginal likelihood (or evidence). The development of the $f$-VI unfolds with a stochastic optimization scheme that utilizes the reparameterization trick, importance weighting and Monte Carlo approximation; a mean-field approximation scheme that generalizes the well-known coordinate ascent variational inference (CAVI) is also proposed for $f$-VI. Empirical examples, including variational autoencoders and Bayesian neural networks, are provided to demonstrate the effectiveness and the wide applicability of $f$-VI.

## 1 Introduction

Variational inference (VI) is a machine learning method that makes Bayesian inference computationally efficient and scalable to large datasets. For decades, the dominant paradigm for approximate Bayesian inference $p(z|x) = p(z, x)/p(x)$ has been Markov-Chain Monte-Carlo (MCMC) algorithms, which estimate the evidence $p(x) = \int p(z, x)dz$ via sampling. However, since sampling tends to be a slow and computationally intensive process, these sampling-based approximate inference methods fade when dealing with the modern probabilistic machine learning problems that usually involve very complex models, high-dimensional feature spaces and large datasets. In these instances, VI becomes a good alternative to perform Bayesian inference. The foundation of VI is primarily optimization rather than sampling. To perform VI, we posit as a family of approximate (or recognition) densities $\mathcal{Q}$ and find the member $q^*(z) \in \mathcal{Q}$ that minimizes the statistical divergence to the true posterior $D(q(z)\|p(z|x))$. Meanwhile, since VI also has many elegant and favorable theoretical properties, *e.g.* variational bounds of the true evidence, it has become the foundation of many popular generative and machine learning models.

Recent advances in VI can be roughly categorized into three groups, improvements over traditional VI algorithms [4, 5], developments of scalable VI methods [6–8], and explorations for tighter variational bounds [9, 10]. Comprehensive reviews on VI's background and progression can be found in [11, 12]. While most of these advancements were built on the classical VI associated with the Kullback–Leibler (KL) divergence, some recent efforts tried to extend the VI framework to other statistical divergences and showed promising results. Among these efforts, Rényi's $\alpha$-divergence and $\chi$-divergence as the *root* divergences (or generators) of the KL divergence were employed for VI in [2, 3, 13], which

---

[*] Authors contributed equally to this paper.

not only broadens the variety of statistical divergences for VI, but makes KL-VI a special case of their methods. Stochastic optimization methods from KL-VI, such as stochastic VI [6] and black-box VI [14], were generalized to Rényi's $\alpha$-VI and $\chi$-VI in [2, 3], and the modified algorithms with new divergences outperformed the classical KL-VI in some benchmarks of Bayesian regressions and image reconstruction. Nevertheless, mean-field approximation, an important type of KL-VI algorithms including the coordinate ascent variational inference (CAVI) and expectation propagation (EP) algorithms [11, 15, 16], were regretfully not revisited or extended for these new divergences.

As the root divergence of the Rényi's $\alpha$-divergence, $\chi$-divergence and many other useful divergences [17, 18], $f$-divergence is a more inclusive statistical divergence (family) and was utilized to improve the statistical properties [19, 20], sharpness [10, 21], and surely the generality of variational bounds [10, 21, 22]. However, most of these works only dealt with some portions of $f$-divergences for their favorable statistical properties, *e.g.* mass-covering [19] and tail-adaptive [20], and did not develop a systematic VI framework that harbors all $f$-divergences. Meanwhile, since i) the regular $f$-divergence does not explicitly induce an $f$-variational bound as elegant as the ELBO [11], $\chi$ upper bound (CUBO) [3], or Rényi variational bound (RVB) [2], and ii) only specific choices of $f$-divergence result in an $f$-variational bound that trivially depends on the evidence [12], a thorough and comprehensive analysis on the $f$-divergence VI has been due for a long time.

In this paper, we extend the traditional VI to $f$-divergence, a rich family that comprises many well-known divergences as special cases [17], by offering some new insights into the $f$-divergence VI and a unified $f$-VI framework that encompasses a number of recent developments in VI methods. An explicit benefit of $f$-VI is that it allows to perform VI or Bayesian approximation with even more variety of divergences, which can potentially bring us sharper variational bounds, more accurate estimate of true evidence, faster convergence rates, more criteria for selecting approximate model $q(z)$, *etc*. We hope this effort can be the last brick to complete the building of $f$-divergence VI and motivate more useful and efficient VI algorithms in the future. After reviewing the $f$-divergence and introducing a crafty surrogate $f$-divergence that is interchangeable with the regular $f$-divergence, we make the following contributions:

$c$1) We enrich the $f$-divergence VI theory by introducing an $f$-VI scheme via minimizing a surrogate $f$-divergence, which makes our $f$-VI framework compatible with the traditional VI approaches and naturally unifies an amount of existing VI methods, such as KL-VI [1], $\alpha$-VI [2], $\chi$-VI [3], and their related developments [7–10, 20].

$c$2) We derive an $f$-variational bound for the evidence and equip it with the upper/lower bound criteria and an importance-weighted (IW-)bound. The $f$-variational bound is realized with an arbitrary $f$-divergence and unifies many existing bounds, such as ELBO, CUBO, RVB, and a number of generalized evidence bounds (GLBO) [10].

$c$3) We propose a universal optimization solution that comprises a stochastic optimization algorithm and a mean-field approximation algorithm for $f$-VI subject to all $f$-divergences, whether or not the $f$-variational bounds trivially depend on the evidence. Experiments on Bayesian neural networks and variational autoencoders (VAEs) show that $f$-VI can be comparable to, or even better than, a number of the state-of-the-art variational methods.

## 2   Preliminary of $f$-divergence

We first introduce some definitions and properties related to $f$-divergence, which are to be adopted in our subsequent exposition.

### 2.1   $f$-divergence

An $f$-divergence that measures the difference between two continuous probability distributions $q$ and $p$ can be defined as follows [17].

**Definition 1** *The $f$-divergence from probability density functions $q(z)$ to $p(z)$ is defined as*

$$D_f(q(z)\|p(z)) =: \int f\left(\frac{q(z)}{p(z)}\right) p(z)\, dz = \mathbb{E}_p\left[f\left(\frac{q(z)}{p(z)}\right)\right], \tag{1}$$

*where $f(\cdot)$ is a convex function with $f(1) = 0$.*

Definition 1 assumes that $q(z)$ is absolutely continuous w.r.t. $p(z)$, which might not be exhaustive, but avoids the unnecessary entanglements with measure theory details. One can however refer to [17, 18]

for a more rigorous treatment. Most prevailing divergences adopted in VI can be regarded as the special cases of $f$-divergence and hence be restored by choosing a proper $f$-function $f(\cdot)$. Table 1 and [17, 18, 21] present the relationship between some well-known statistical divergences adopted in VI and their $f$-functions. Intuitively, one can perform $f$-VI by minimizing either the *forward* $f$-divergence $D_f(p\|q)$ or the *reverse* $f$-divergence $D_f(q\|p)$, and [21, 23] provide some heuristic discussions on their statistical differences. Since VI based on the reverse KL divergence is more tractable to compute and more statistically sensible, we will develop our $f$-VI framework primarily based on the reverse $f$-divergence, while one can still unify or commute between the forward and reverse $f$-divergences via the *dual function* $f^*$, which is also referred to as the perspective function or the conjugate symmetry of $f$ in [3, 17, 24].

**Definition 2** *Given a function $f : (0, \infty) \to \mathbb{R}$, the dual function $f^* : (0, \infty) \to \mathbb{R}$ is defined as*
$$f^*(t) = t \cdot f(1/t).$$

One can verify that the dual function $f^*$ has the following properties: i) $(f^*)^* = f$; ii) if $f$ is convex, $f^*$ is also convex, and iii) if $f(1) = 0$, then $f^*(1) = 0$. With dual function $f^*$, an identity between the forward and reverse $f$-divergences can be established [3]:
$$D_{f^*}(p\|q) = \int \frac{p(z)}{q(z)} \cdot f\left(\frac{q(z)}{p(z)}\right) \cdot q(z)\, dz = D_f(q\|p).$$

In order to facilitate the derivation of $f$-variational bound, especially when the latent variable model is involved [21, 25], we introduce a *surrogate $f$-divergence* $D_{f_\lambda}$ defined by the *generator function*
$$f_\lambda(\cdot) = f(\lambda\cdot) - f(\lambda), \tag{2}$$
where $\lambda \geq 0$ is constant. It is straightforward to verify that $f$ and $f_\lambda$ have the same convexity, and $f(1) = 0$ implies $f_\lambda(1) = 0$, which induces a valid (surrogate) $f$-divergence, denoted as $D_{f_\lambda}$, that can virtually replace $D_f$ when needed[2]. To justify the closeness between divergences $D_f$ and $D_{f_\lambda}$, we first note that $D_f$ and $D_{f_\lambda}$ share the same minimum point at $p = q$, then we have the following statement.

**Proposition 1** *Given two probability distributions $q$ and $p$, a convergent sequence $\lim_{n\to\infty} \lambda_n = 1, \lambda_n \geq 0$, and a convex function $f : (0, +\infty) \to \mathbb{R}$ such that $f(1) = 0$ and $f(\cdot)$ is uniformly continuous, the $f$-divergences between $q$ and $p$ satisfy*
$$D_{f_{\lambda_n}}(q\|p) \to D_f(q\|p) \tag{3}$$
*almost everywhere as $n \to \infty$.*

## 2.2 Shifted homogeneity

We then introduce a class of $f$-functions equipped with a structural advantage in decomposition, which will be invoked later to derive the coordinate-wise VI algorithm under mean-field assumption.

**Definition 3** *A convex function $f$ belongs to $\mathcal{F}_{\{0,1\}}$, if $f(1) = 0$, and for any $t, \tilde{t} \in \mathbb{R}$, we have*
$$f(t\tilde{t}) = t^\gamma f(\tilde{t}) + f(t)\tilde{t}^\eta, \tag{4}$$
*where $\gamma \in \mathbb{R}$, and $\eta \in \{0, 1\}$. Function $f$ is type $0$ shifted homogeneous or $f \in \mathcal{F}_0$ if $\eta = 0$, and type $1$ shifted homogeneous or $f \in \mathcal{F}_1$ if $\eta = 1$.*

This special class of functions allows to decompose an $f$-function into two or more (by iterations) terms, each of which is a product of an $f$-function and an exponent. In Table 1, we show that the $f$-functions of many well-known divergences can be classified as $\mathcal{F}_{\{0,1\}}$ functions.

Table 1: Divergences $D_f(q\|p)$ and homogeneity decomposition.

| Divergences | $f(t)$ | $f(t\tilde{t})$ |
|---|---|---|
| KL divergence [1] | $t \log t$ | $t f(\tilde{t}) + f(t)\tilde{t}$ |
| General $\chi^n$-divergence [3] | $t^n - 1, n \in \mathbb{R}\backslash(0,1)$ | $t^n f(\tilde{t}) + f(t)$ |
| Hellinger $\alpha$-divergence $\mathcal{H}_\alpha$ [18] | $(t^\alpha - 1)/(\alpha - 1), \alpha \in \mathbb{R}^+\backslash\{1\}$ | $t^\alpha f(\tilde{t}) + f(t)$ |
| Rényi's $\alpha$-divergence[3] [2] | $D_\alpha(q\|p) = (\alpha - 1)^{-1} \log[1 + (\alpha - 1)\mathcal{H}_\alpha(q\|p)]$ | |

The duality property between $\mathcal{F}_0$ and $\mathcal{F}_1$ is stated in Proposition 2.

**Proposition 2** *Given $f_0 \in \mathcal{F}_0$ and $f_1 \in \mathcal{F}_1$, the dual functions $f_0^* \in \mathcal{F}_1$ and $f_1^* \in \mathcal{F}_0$.*

When $f \in \mathcal{F}_{\{0,1\}}$, we can establish a more profound relationship, in contrast with Proposition 1, between $f$-divergence $D_f$ and surrogate divergence $D_{f_\lambda}$.

**Proposition 3** *When $f \in \mathcal{F}_{\{0,1\}}$ and $\lambda > 0$, an $f$-divergence $D_f$ and its surrogate divergence $D_{f_\lambda}$ satisfy*

$$D_{f_\lambda}(q\|p) = \lambda^\gamma D_f(q\|p). \tag{5}$$

By virtue of the equivalence relationship revealed in Proposition 1 and 3, we can interchangeably use $f$-divergence $D_f$ and surrogate divergence $D_{f_\lambda}$, and the parameter $\lambda$ of surrogate divergence provides an additional degree of freedom when deriving the variational bounds and VI algorithms.

## 3 Variational bounds and optimization

While it was difficult to retrieve an $f$-variational bound [10, 20, 21], which is an expectation over $q$ and unifies the existing variational bounds [2, 3, 11], by directly manipulating the original $f$-divergence in (1), we will show that such a general variational bound can be found when minimizing a crafty surrogate $f$-divergence.

### 3.1 $f$-variational bounds

Given a convex function $f$ such that $f(1) = 0$ and a set of i.i.d. samples $\mathcal{D} = \{x^{(n)}\}_{n=1}^N$, the generator function $f_{p(\mathcal{D})^{-1}}$ with $p(\mathcal{D}) > 0$ can induce a surrogate $f$-divergence. Our $f$-VI is then initiated from minimizing the following reverse (surrogate) $f$-divergence

$$D_{f_{p(\mathcal{D})^{-1}}}(q(z)\|p(z|\mathcal{D})) = \frac{1}{p(\mathcal{D})} \cdot \mathbb{E}_{q(z)}\left[f^*\left(\frac{p(z,\mathcal{D})}{q(z)}\right)\right] - f\left(\frac{1}{p(\mathcal{D})}\right). \tag{6}$$

Multiplying both sides of (6) by $p(\mathcal{D})$ and with rearrangements, we have

$$\mathcal{L}_f(q, \mathcal{D}) = \mathbb{E}_{q(z)}\left[f^*\left(\frac{p(z,\mathcal{D})}{q(z)}\right)\right] = f^*(p(\mathcal{D})) + p(\mathcal{D}) \cdot D_{p(\mathcal{D})^{-1}}(q(z)\|p(z|\mathcal{D})). \tag{7}$$

For a given evidence $p(\mathcal{D})$, we can minimize the $f$-divergence $D_{f_{p(\mathcal{D})^{-1}}}(q(z)\|p(z|\mathcal{D}))$ by minimizing the expectation in (7), which is defined as the $f$-variational bound $\mathcal{L}_f(q, \mathcal{D})$. Consequently, by the non-negativity of $f$-divergence [17, 18], we can establish the following inequality.

**Theorem 1** *Dual function of evidence $f^*(p(\mathcal{D}))$ is bounded above by $f$-variational bound $\mathcal{L}_f(q, \mathcal{D})$*

$$\mathcal{L}_f(q, \mathcal{D}) = \mathbb{E}_{q(z)}\left[f^*\left(\frac{p(z,\mathcal{D})}{q(z)}\right)\right] \geq f^*(p(\mathcal{D})), \tag{8}$$

*and equality is attained when $q(z) = p(z|\mathcal{D})$, i.e. $D_{p(\mathcal{D})^{-1}}(q(z)\|p(z|\mathcal{D})) = 0$.*[4]

By properly choosing $f$-function, $f$-variational bound $\mathcal{L}_f(q, \mathcal{D})$ and (8) can restore the most existing variational bounds and the corresponding inequalities, *e.g.* $f(t) = t\log(t)$ for ELBO in [11] and $f(t) = t^{1-n} - t$ for CUBO in [3]. See Supplementary Material (SM) for more restoration examples and some new variational bounds, *e.g.* an evidence upper bound (EUBO) under KL divergence. While the assumption of $p(\mathcal{D}) > 0$ or the existence of $p(\mathcal{D})^{-1}$ in (6) might lay additional restrictions in some situations, we can circumvent them by resorting to the $f$-VI minimizing the forward surrogate $f$-divergence $D_{f_{p(\mathcal{D})}}(p(z|\mathcal{D})\|q(z))$. SM provides more details for this alternative. Additionally, $\mathcal{L}_f(q, \mathcal{D})$ in (8) can be further sharpened by leveraging multiply-weighted posterior samples [9], *i.e.*, importance-weighted VI.

$$\mathbb{E}_{q(z)}\left[f^*\left(\frac{p(z,\mathcal{D})}{q(z)}\right)\right] \geq f^*\left(\mathbb{E}_{q(z)}\left[\frac{p(z,\mathcal{D})}{q(z)}\right]\right) = f^*(p(\mathcal{D})).$$

**Corollary 1** *When $1 \leq L_1 \leq L_2$, the importance-weighted $f$-variational bound $\mathcal{L}_f^{\text{IW}}(q, \mathcal{D}, L)$ and the $f$-variational bound $\mathcal{L}_f(q, \mathcal{D})$ satisfy*

$$\mathcal{L}_f(q, \mathcal{D}) \geq \mathcal{L}_f^{\text{IW}}(q, \mathcal{D}, L_1) \geq \mathcal{L}_f^{\text{IW}}(q, \mathcal{D}, L_2) \xrightarrow{L \to \infty} f^*(p(\mathcal{D})),$$

*where $\mathcal{L}_f^{\text{IW}}(q, \mathcal{D}, L)$ is defined as*

$$\mathcal{L}_f^{\text{IW}}(q, \mathcal{D}, L) = \mathbb{E}_{z_{1:L} \sim q(z)} \left[ f^* \left( \frac{1}{L} \sum_{l=1}^{L} \frac{p(z_l, \mathcal{D})}{q(z_l)} \right) \right],$$

*and $z_{1:L} = \{z_l\}_{l=1}^{L}$ are $L \in \mathbb{N}^*$ i.i.d. samples from $q(z)$.*

For clarity and conciseness, we will develop the subsequent results primarily based on $\mathcal{L}_f(q, \mathcal{D})$. Nevertheless, our readers should feel safe to replace $\mathcal{L}_f(q, \mathcal{D})$ with $\mathcal{L}_f^{\text{IW}}(q, \mathcal{D}, L)$ in the following context and obtain improved outcomes. More interesting results can be observed from (8). After composing both sides of (8) with the inverse dual function $(f^*)^{-1}$, we have the following observations:

*o*1) When the dual function $f^*$ is increasing (or non-decreasing) on $\mathbb{R}^+$, the composition gives an evidence upper bound:
$$(f^*)^{-1} \circ \mathcal{L}_f(q, \mathcal{D}) \geq p(\mathcal{D}).$$

*o*2) When the dual function $f^*$ is decreasing (or non-increasing) on $\mathbb{R}^+$, the composition gives an evidence lower bound:
$$(f^*)^{-1} \circ \mathcal{L}_f(q, \mathcal{D}) \leq p(\mathcal{D}).$$

*o*3) When the dual function $f^*$ is non-monotonic on $\mathbb{R}^+$, the composition gives a local evidence bound by applying *o*1) or *o*2) on a monotonic interval of $f^*$.

Based on these observations, we can readily imply a sandwich formula for evidence $p(\mathcal{D})$, which is essential for accurate VI [12].

**Corollary 2** *Given convex functions $f$ and $g$ such that $f(1) = g(1) = 0$, on an interval where $f^*$ is increasing and $g^*$ is decreasing, the evidence $p(\mathcal{D})$ satisfies*

$$(g^*)^{-1} \circ \mathbb{E}_{q(z)} \left[ g^* \left( \frac{p(z, \mathcal{D})}{q(z)} \right) \right] \leq p(\mathcal{D}) \leq (f^*)^{-1} \circ \mathbb{E}_{q(z)} \left[ f^* \left( \frac{p(z, \mathcal{D})}{q(z)} \right) \right]. \qquad (9)$$

The evidence bounds in (9) are akin to the GLBO, which was proposed on the basis of a few assumptions and intuitions in [10]. Corollary 1 and Corollary 2 interprets and supplements GLBO with rigorous $f$-VI analysis and explicit instructions on choosing $f$-function. Compared with the unilateral variational bounds, the bilateral bounds in (9) reveal more information and allow to estimate $p(\mathcal{D})$ with more accuracy. To sharpen these bilateral bounds, we need to properly choose the functions $f$ and $g$ and the recognition model $q(z)$ such that $\sup_{g,q} g^{-1} \circ \mathcal{L}_g(q, \mathcal{D})$ and $\inf_{f,q} f^{-1} \circ \mathcal{L}_f(q, \mathcal{D})$ can be attained. For a selected family of $q(z)$, various choices of $f$ and $g$ will lead to evidence bounds of different sharpness and optimization efficiency. The model selection of approximate distribution $q(z)$ is a fundamental problem inherited by all VI algorithms, and a feasible solution is to compare the performance of candidate models while fixing an $f$- or $g$-function [10] or alternating among some common divergences. Once the functions $f$ and $g$ and the recognition model $q(z)$ are determined, we can approximate the optimal distribution $q^*(z)$ in $q(z)$ or minimize $\mathcal{L}_f(q, \mathcal{D})$ by adjusting the parameters in $q(z)$, which does not require the dual function $f^*$ or $g^*$ be invertible as in (9) and will be discussed in the succeeding subsections.

## 3.2 Stochastic optimization

While classical VI is limited to conditionally conjugate exponential family models [11, 12, 23], the stochastic optimization makes VI applicable for more modern and complicated problems [6, 14]. To minimize $\mathcal{L}_f(q, \mathcal{D})$ with stochastic optimization, we supplement the preceding VI formulation with more details. The approximate model is formulated as $q_\theta(z)$, where $\theta \in \mathbb{R}^M$ are the parameters to be optimized. While some papers [7, 10, 26] also consider and optimize the parameters $\phi$ in the generative model $p_\phi$, we prefer to treat the parameters $\phi$ as latent variables $z$ for conciseness. An

intuitive approach to apply stochastic optimization is to compute the standard gradient of $\mathcal{L}_f(q, \mathcal{D})$ or $\mathcal{L}_f^{\text{IW}}(q, \mathcal{D})$ w.r.t. $\theta$

$$\nabla_\theta \mathcal{L}_f(q_\theta, \mathcal{D}) = \mathbb{E}_{q_\theta(z)} \left[ f' \left( \frac{q_\theta(z)}{p(z, \mathcal{D})} \right) \cdot \nabla_\theta \log q_\theta(z) \right], \tag{10}$$

where $f'(t)$ denotes $\partial f(t)/\partial t$. Since $\nabla_\theta \log q_\theta(z)$ is known as the score function in statistics [27] and is a part of the REINFORCE algorithm [26, 28], (10) is called score function or REINFORCE gradient. An unbiased Monte Carlo (MC) estimator for (10) can be obtained by drawing $z_1, z_2, \cdots, z_K$ from $q_\theta(z)$ and

$$\nabla_\theta \hat{\mathcal{L}}_f(q_\theta, \mathcal{D}) = \frac{1}{K} \sum_{k=1}^{K} \left[ f' \left( \frac{q_\theta(z_k)}{p(z_k, \mathcal{D})} \right) \cdot \nabla_\theta \log q_\theta(z_k) \right]. \tag{11}$$

However, since the variance of estimator (11) can be too large to be useful in practice, the score function gradient is usually employed along with some variation reduction techniques, such as the control variates and Rao-Blackwellization [14, 26, 29].

An alternative to the score function gradient is the reparameterization gradient, which empirically has a lower estimation variance [7, 21] and can be integrated with neural networks. The reparameterization trick requires the existence of a noise variable $\varepsilon \sim p(\varepsilon)$ and a mapping $g_\theta(\cdot)$ such that $z = g_\theta(\varepsilon)$. Instead of directly sampling $\{z_k\}_{k=1}^K$ from $q_\theta(z)$, the reparameterization estimators rely on the samples $\{\varepsilon_k\}_{k=1}^K$ drawn from $p(\varepsilon)$, for example, a Gaussian latent variable $z \sim q_\theta(z) = \mathcal{N}(\mu, \Sigma)$ can be reparameterized with a standard Gaussian variable $\varepsilon \sim \mathcal{N}(0, 1)$ and a mapping $z = g_\theta(\varepsilon) = \mu + \Sigma^{\frac{1}{2}} \varepsilon$. More detailed interpretations as well as recent advances in the reparameterization trick can be found in [7, 30–32]. The gradient of $\mathcal{L}_f(q, \mathcal{D})$ after reparameterization becomes

$$\nabla_\theta \mathcal{L}_f^{\text{rep}}(q_\theta, \mathcal{D}) = \nabla_\theta \mathbb{E}_{p(\varepsilon)} \left[ f^* \left( \frac{p(g_\theta(\varepsilon), \mathcal{D})}{q_\theta(g_\theta(\varepsilon))} \right) \right]. \tag{12}$$

An unbiased MC estimator for (12) is

$$\nabla_\theta \hat{\mathcal{L}}_f^{\text{rep}}(q_\theta, \mathcal{D}) = \frac{1}{K} \sum_{k=1}^{K} \nabla_\theta f^* \left( \frac{p(g_\theta(\varepsilon_k), \mathcal{D})}{q_\theta(g_\theta(\varepsilon_k))} \right), \tag{13}$$

where $\varepsilon_1, \varepsilon_2, \cdots, \varepsilon_K$ are drawn from $p(\varepsilon)$. We also give an unbiased MC estimator for the importance-weighted reparameterization gradient in (14), which will be utilized in later experiments:

$$\nabla_\theta \hat{\mathcal{L}}_f^{\text{IW, rep}}(q_\theta, \mathcal{D}, L) = \frac{1}{K} \sum_{k=1}^{K} \nabla_\theta f^* \left( \frac{1}{L} \sum_{l=1}^{L} \frac{p(g_\theta(\varepsilon_{k,l}), \mathcal{D})}{q_\theta(g_\theta(\varepsilon_{k,l}))} \right), \tag{14}$$

where noise samples $\{\varepsilon_{k,1:L}\}_{k=1}^K$ are drawn from $p(\varepsilon)$. All the aforementioned estimators for $f$-variational bounds and gradients are unbiased, while composing these estimator with other functions, *e.g.* inverse dual functions in (9), can sometimes trade the unbiasedness for numerical stability [2, 3, 10].

Nonetheless, the preceding estimators and VI algorithms rely on the full dataset $\mathcal{D}$ and can be handicapped to tackle the problems with large datasets. Meanwhile, since the properties of $f^*$-functions are flexible, it is non-trivial to represent the $f$-variational bounds by the expectation on a datapoint-wise loss, except for some specific divergences, such as KL divergence [7] or divergences with dual functions $f^*$ satisfying $f^*(t\tilde{t}) = f^*(t) + f^*(\tilde{t})$, *i.e.* $f^* \in \mathcal{F}_0$ with $\gamma = 0$. Therefore, to deploy the mini-batch training, we integrate the aforementioned estimators with the *average likelihood* technique [2]. Given a mini-batch of $M$ datapoints $\mathcal{D}_M = \{x_{n1}, \cdots, x_{nM}\} \subset \mathcal{D}$, we approximate the full log-likelihood by $\log p(\mathcal{D}|z) \approx N/M \cdot \sum_{m=1}^{M} \log p(x_{nm}|z)$. Hence, the ratio $p(z, \mathcal{D})/q(z)$ in (10-14) can be approximated by $\log[p(z, \mathcal{D})/q(z)] \approx N/M \cdot \sum_{m=1}^{M} \log p(x_{nm}|z) + \log p(z) - \log q(z)$. When $z$ contains local hidden variables, the prior distribution $p(z)$ and approximate distribution $q(z)$ should also be approximated accordingly. This proxy to the full dataset wraps up our black-box $f$-VI algorithm, which is essentially a stochastic optimization algorithm that only relies on a mini-batch of data in each iteration. A reference black-box $f$-VI algorithm and the optimization schemes for a few concrete divergences are given in the SM.

## 3.3 Mean-field approximation

Mean-field approximation, which simplifies the original VI problem for tractable computation, is historically an important VI algorithm before the emergence of stochastic VI. As the cornerstone of several variational message passing algorithms [33, 34], mean-field VI is still evolving [4, 5, 11, 12] and worthy to be generalized for $f$-VI. A mean-field approximation assumes that all latent variables $\{z_j\}_{j=1}^J$ are independent, and the recognition model can be fully factorized as $q(z) = \prod_{j=1}^J q_j(z_j)$, which simplifies the derivations and computation but might lead to less accurate results. The mean-field $f$-VI algorithm alternatively updates each marginal distribution $q_j$ to minimize the $f$-variational bound $\mathcal{L}_f(q, \mathcal{D})$. For the $f$-divergences with $f \in \mathcal{F}_1$, such as KL divergence, the coordinate-wise update rule for $q_j(z_j)$ is obtained from fixing the other variational factors $q_{-j}(z_{-j}) = \prod_{\ell \neq j} q_\ell(z_\ell)$ and singling out $q_j(z_j)$ from $f$-variational bound $\mathcal{L}_f(q, \mathcal{D})$ in (8), which gives

$$q_j^*(z_j) \propto f^{*-1}\left( \mathbb{E}_{q_{-j}} \left[ f^* \left( \frac{p(z, \mathcal{D})}{q_{-j}(z_{-j})} \right) \right] \right). \tag{15}$$

For the $f$-divergences with $f \in \mathcal{F}_0$, such as $\chi$- or Rényi's $\alpha$-divergences, the coordinate-wise update rule for $q_j(z_j)$ is obtained by applying the same procedures to the $f$-variational bound $\mathcal{L}_f(q, \mathcal{D}) = \mathbb{E}_{q(z)}[f(p(z, \mathcal{D})/q(z))]$ from the forward $f$-VI (see SM), which gives

$$q_j^*(z_j) \propto f^{-1}\left( \mathbb{E}_{q_{-j}} \left[ f \left( \frac{p(z, \mathcal{D})}{q_{-j}(z_{-j})} \right) \right] \right). \tag{16}$$

When deriving these mean-field $f$-VI update rules (see SM), we only exploit the homogeneity of $f$- or $f^*$-function. CAVI [11, 15], EP [16], and other types of mean-field VI algorithms can be restored from (15) and (16) by choosing a proper $f$- or $f^*$-function. A reference mean-field VI algorithm along with a concrete realization example under KL divergence is provided in the SM. When the inverse function $f^{*-1}$ or $f^{-1}$ in (15) or (16) is not analytically solvable, we can either generate a lookup table for $f^{*-1}$ or $f^{-1}$ and numerically evaluate (15) or (16) or resort to the stochastic $f$-VI.

## 4 Experiments

The effectiveness and the wide applicability of $f$-VI are demonstrated with three empirical examples in this section. We first verify the theoretical results with a synthetic example. The $f$-VI is then respectively implemented for a Bayesian neural network for linear regression and a VAE for image reconstruction and generation. Adam optimizer with recommended parameters in [35] is employed for stochastic optimization, if not specified. Empirical results and data are reported by their mean value and 95% confidence intervals. More detailed descriptions on the experimental settings, supplemental results, and the demonstration of the mean-field approximation method are provided in the SM.

### 4.1 Synthetic example

We first demonstrate the $f$-VI theory with a vanilla example. Consider a batch of i.i.d. datapoints generated by a latent variable model $x = \sin(z) + \mathcal{N}(0, 0.01)$, $z \sim \text{UNIF}(0, \pi)$, where $\mathcal{N}(\mu, \sigma^2)$ denotes a univariate normal distribution with mean $\mu$ and variance $\sigma^2$, and $\text{UNIF}(a, b)$ denotes a uniform distribution on interval $[a, b]$. Subsequently, for simplicity, we posit a prior distribution $p(z) = \text{UNIF}(0, \pi)$, a likelihood distribution $p(x|z) = \mathcal{N}(\sin(z), 0.01)$, and an approximate model $q_\theta(z) = \text{UNIF}(\frac{1-\theta}{2}\pi, \frac{\theta+1}{2}\pi)$, which is a uniform distribution centered at $z = \pi/2$ with width $\theta\pi$. To verify the order and the sharpness of $f$-variational bounds, we fix $\theta = 1.1$ and approximate the true evidence $p(x)$, IW-RVB ($\alpha = 2$), (IW-)CUBO ($n = 2$), and (IW-)ELBO ($L = 8$) in Figure 1(a), which substantiates Theorem 1, Corollary 1 and 2. A variational bound associated with the total variation distance, an $f$-divergence with non-monotonic $f^*$ function, is analyzed in the SM, and more approximation results when $q(z) = \mathcal{N}(\pi/2, 1)$ can be found in [10]. To demonstrate the effectiveness of stochastic $f$-VI algorithm, we set an initial value $\theta_0 = 1.5$ and update the recognition distribution $q_\theta(z)$ by optimizing the IW-RVB ($\alpha = 3$), (IW-)CUBO ($n = 2$), and (IW-)ELBO. The IW-reparameterization gradient (14) with $L = 3$ and $K = 1000$ is adopted for the training on a dataset of 500 observations, and the $f$-variational bounds in Figure 1(b) are evaluated on a test set of 50 observations. The sandwich-type bounds in Figure 1(b) give an estimate of the test log-evidence, which is roughly between $-235$ and $-300$.

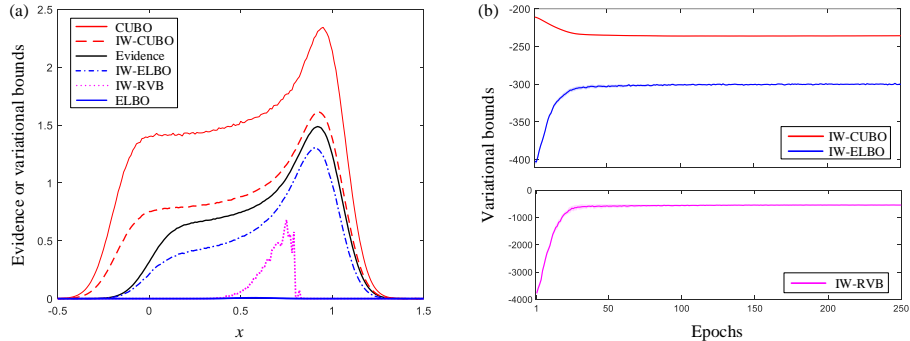

Figure 1: $f$-variational bounds on synthetic data.

## 4.2 Bayesian neural network

We then implement the $f$-VI for a single-layer neural network for Bayesian linear regression. Our experimental setup generally follows the regression settings in [2], while some parameters vary to adapt to the $f$-VI framework. The linear regression is performed with twelve datasets from the UCI Machine Learning Repository [36]. Each dataset is randomly split into $90\%/10\%$ for training and testing, and six different dual functions $f^*(\cdot)$ in $\mathcal{L}_f^{\text{IW}}(q, \mathcal{D}, L)$ are selected such that three well-established $f$-VIs (KL-VI, Rényi's $\alpha$-VI with $\alpha = 3$, and $\chi$-VI with $n = 2$) and three new $f$-VIs (VIs subject to total variation distance and two custom $f$-divergences) are tested and compared. One of the custom $f$-divergences, inspired by [19], is defined by a convex dual function $f_{c1}^*(t) = \tilde{f}^*(t) - \tilde{f}^*(1)$, where $\tilde{f}^*(t) = -1/6 \cdot (\log t + t_0)^3 - 1/2 \cdot (\log t + t_0)^2 - (\log t + t_0) - 1$, $t = p(z, \mathcal{D})/q(z)$, and $t_0 \in \mathbb{R}$ is a parameter to be optimized. The IW-reparameterization gradient with $L = 5$, $K = 50$ and mini-batch size of 32 is employed for training. After 20 trials with 500 training epochs in each trial, the regression results are evaluated by the test root mean squared error (RMSE) and test negative log-likelihood reported in Table 2. The performance of custom $f_{c1}$-VI matches the results of well-established $f$-VIs on most datasets, and the custom $f_{c1}$-VI quantitatively outperforms others on some datasets, *e.g.* Fish Toxicity and Stock. A complete version of Table 2, including the regression results of the other two new $f$-VIs, and more detailed descriptions on the training process, such as the architecture of neural network, training parameters, numerical stability and estimator biasedness, are provided in the SM.

Table 2: Average test error and negative log likelihood.

| Dataset | Test RMSE (lower is better) | | | | Test negative log-likelihood (lower is better) | | | |
|---|---|---|---|---|---|---|---|---|
| | KL-VI | $\chi$-VI | $\alpha$-VI | $f_{c1}$-VI | KL-VI | $\chi$-VI | $\alpha$-VI | $f_{c1}$-VI |
| Airfoil | **2.16**±**.07** | 2.36±.14 | 2.30±.08 | 2.34±.09 | **2.17**±**.03** | 2.27±.03 | 2.26±.02 | 2.29±.02 |
| Aquatic | **1.12**±**.06** | 1.20±.06 | 1.14±.07 | 1.14±.06 | **1.54**±**.04** | 1.60±.08 | 1.54±.07 | 1.54±.06 |
| Boston | **2.76**±**.36** | 2.99±.37 | 2.86±.36 | 2.87±.36 | 2.49±.08 | 2.54±.18 | **2.48**±**.13** | 2.49±.13 |
| Building | 1.38±.12 | 2.82±.51 | 1.83±.22 | 1.80±.21 | 6.62±.02 | 6.94±.13 | 6.79±.03 | 6.74±.04 |
| CCPP | **4.05**±**.09** | 4.14±.11 | 4.06±.08 | 4.33±.12 | **2.82**±**.02** | 2.84±.03 | 2.82±.02 | 2.95±.01 |
| Concrete | 5.40±.24 | **3.32**±**.34** | 5.32±.27 | 5.26±.21 | 3.10±.04 | **2.61**±**.18** | 3.09±.04 | 3.09±.03 |
| Fish Toxicity | 0.88±.04 | 0.90±.04 | 0.89±.04 | 0.88±.03 | 1.28±.04 | 1.27±.04 | 1.29±.04 | 1.29±.03 |
| Protein | 1.93±.19 | 2.45±.42 | **1.87**±**.17** | 1.97±.21 | **2.00**±**.07** | 2.01±.08 | 2.04±.08 | 2.21±.04 |
| Real Estate | 7.48±1.41 | 7.51±1.44 | **7.46**±**1.42** | 7.52±1.40 | 3.60±.30 | 3.70±.45 | **3.59**±**.32** | 3.62±.33 |
| Stock | 3.85±1.12 | 3.90±1.09 | 3.88±1.13 | **3.82**±**1.11** | -1.09±.04 | -1.09±.04 | -1.09±.04 | -1.09±.04 |
| Wine | .642±.018 | .640±.021 | .638±.018 | .643±.019 | .966±.027 | .965±.028 | .964±.025 | .975±.027 |
| Yacht | **0.78**±**.12** | 1.18±.18 | 0.99±.12 | 1.00±.18 | **1.70**±**.02** | 1.79±.03 | 1.82±.01 | 2.05±.01 |

## 4.3 Bayesian variational autoencoder

We also integrate the $f$-VI with a Bayesian VAE for image reconstruction and generation on the datasets of Caltech 101 Silhouettes [37], Frey Face [38], MNIST [39], and Omniglot [40]. By replacing the conventional ELBO loss function of VAE [7, 41] with the more flexible $f$-variational bound loss functions, we test and compare the $f$-VAEs associated with three well-known $f$-divergences (KL-divergence, Rényi's $\alpha$-divergence with $\alpha = 3$, and $\chi$-divergence with $n = 2$) and three new

$f$-divergences (total variation distance and two custom $f$-divergences). The dual function for total variation distance is $f^*(t) = |t - 1|$. The custom $f_{c1}$-variational bound loss is induced by the aforementioned dual function $f_{c1}^*(t) = \tilde{f}^*(t) - \tilde{f}^*(1)$ with $t_0 = 0$. The custom $f_{c2}$-variational bound loss is induced by dual function $f_{c2}^*(t) = \log^2 t + \log t$, which is convex on $t = p(z, \mathcal{D})/q(z) \in (0, 1)$. The reparameterization gradient with $K = 3$, $L = 1$ is used for training. After 20 trials with 200 training epochs in each trial, the average test reconstruction errors (lower is better) measured by cross-entropy are listed in Table 3. In $f$-VAE example, the performances of three new $f$-VIs also rival the results of three well-known $f$-VIs on most datasets. Reconstructed and generated images, architectures of the encoder and decoder networks, and more detailed interpretations on the custom $f$-functions and training process of $f$-VAEs are given in the SM.

Table 3: Average test reconstruction errors of $f$-VAEs.

|  | KL-VI | $\chi$-VI | $\alpha$-VI | TV-VI | $f_{c1}$-VI | $f_{c2}$-VI |
|---|---|---|---|---|---|---|
| Caltech 101 | **73.80**±**2.27** | 73.84±2.16 | 74.95±2.76 | 74.32±2.26 | 74.87±2.56 | 74.85±2.94 |
| Frey Face | 160.85±.72 | 160.57±.95 | 161.06±1.16 | 161.11±1.00 | **160.52**±**.88** | 160.65±.87 |
| MNIST | **59.06**±**.40** | 62.13±.50 | 61.90±.69 | 62.44±.41 | 59.60±.25 | 59.53±.42 |
| Omniglot | 109.62±.20 | 110.57±.28 | 110.81±.32 | 110.21±.31 | **107.13**±**.39** | 108.29±.28 |

## 5  Conclusion

We have introduced a general $f$-divergence VI framework equipped with a rigorous theoretical analysis and a standardized optimization solution, which together extend the current VI methods to a broader range of statistical divergences. Empirical experiments on the popular benchmarks imply that this $f$-VI method is flexible, effective, and widely applicable, and some custom $f$-VI instances can attain state-of-the art results. Future work on $f$-VI may include finding the $f$-VI instances with more favorable properties, more efficient $f$-VI optimization methods, and VI frameworks and theories that are more universal than the $f$-VI.

## Broader Impact

This work does not present any foreseeable societal consequence.

## Acknowledgments and Disclosure of Funding

This work was supported by AFSOR under Grant FA9550-15-1-0518 and NSF NRI under Grant ECCS-1830639. The authors would like to thank the anonymous editors and reviewers for their constructive comments, Dr. Xinyue Chang (Iowa State Univ.), Lei Ding (Univ. of Alberta), Zhaobin Kuang (Stanford), Yang Wang (Univ. of Alabama), and Yanbo Xu (Georgia Tech.) for their helpful suggestions, and Prof. Evangelos A. Theodorou for his heuristic and insightful comments on this paper.

## Footnotes

[2]Essentially, $D_{f_\lambda}$ is an $f$-divergence between a positive measure $\mathbb{P}(\cdot, \lambda)$ and a probability measure $\mathbb{Q}(\cdot)$.

[3]Renyi's $\alpha$-divergence cannot be directly restored from $f$-divergence (1), while it is a one-to-one transformation of $\mathcal{H}_\alpha$ of the same order $\alpha \in \mathbb{R}^+\backslash\{1\}$.

[4]Inequality (8) can also be derived by resorting to Jensen's inequality. Since $f^*$ is convex, we have

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
