[Supplementary Material]

# Supplementary Material for
# "$f$-Divergence Variational Inference"

**Neng Wan**[1]$^*$      **Dapeng Li**[2]$^*$      **Naira Hovakimyan**[1]
nengwan2@illinois.edu    dapeng.ustc@gmail.com    nhovakim@illinois.edu
[1] University of Illinois at Urbana-Champaign, Urbana, IL 61801
[2] Anker Innovations, Shenzhen, China

This supplementary material provides additional details for some results in the original paper.

## A    Proofs of the main results

This section provides i) elaboration on the surrogate $f$-divergence including the proofs of Proposition 1, Proposition 2 and Proposition 3, ii) deviations of the $f$-variational bound generated from both the reverse and forward surrogate $f$-divergence, and iii) an importance-weighted $f$-variational bound and the proof of Corollary 1.

### A.1    Proof of Proposition 1

We first expand the LHS of (3) by substituting the definitions of $f$-divergence (1) and generator function (2).

$$
\begin{aligned}
\lim_{n \to \infty} D_{f_{\lambda_n}}(q\|p) &= \lim_{n \to \infty} \int p(z) \cdot \left[ f\left( \lambda_n \cdot \frac{q(z)}{p(z)} \right) - f(\lambda_n) \right] \, dz \\
&= \lim_{n \to \infty} \int p(z) \cdot f\left( \lambda_n \cdot \frac{q(z)}{p(z)} \right) \, dz - \lim_{n \to \infty} f(\lambda_n) \cdot \int p(z) \, dz \\
&= \lim_{n \to \infty} \int p(z) \cdot f\left( \lambda_n \cdot \frac{q(z)}{p(z)} \right) \, dz.
\end{aligned}
$$

In order to prove (3), we only need to show that

$$
\lim_{n \to \infty} \int p(z) \cdot f\left( \lambda_n \cdot \frac{q(z)}{p(z)} \right) \, dz = \int \lim_{n \to \infty} p(z) \cdot f\left( \lambda_n \cdot \frac{q(z)}{p(z)} \right) \, dz = D_f(q\|p), \qquad (17)
$$

which can be proved by showing that function $g(\lambda) = \int p(x) \cdot f\left( \lambda \cdot q(z)/p(z) \right) dz$ is continuous in $\lambda$, since the continuity of $g(\lambda)$ brings each convergent sequence in $\lambda$ to a convergent sequence in $g(\cdot)$. The continuity of $g(\lambda)$ can be justified as follows. For arbitrary $\varepsilon > 0$ and $z$, there exists $\delta$ such that

$$
\begin{aligned}
|g(\lambda + \delta) - g(\lambda)| &= \left| \int p(z) \cdot \left[ f\left( (\lambda + \delta) \cdot \frac{q(z)}{p(z)} \right) - f\left( \lambda \cdot \frac{q(z)}{p(z)} \right) \right] dz \right| \\
&\leq \int p(z) \cdot \left| f\left( (\lambda + \delta) \cdot \frac{q(z)}{p(z)} \right) - f\left( \lambda \cdot \frac{q(z)}{p(z)} \right) \right| dz \\
&\leq \int p(z) \cdot \epsilon \, dz = \varepsilon \,,
\end{aligned}
$$

where we have used the uniform continuity of $f(\cdot)$. This completes the proof.    ∎

---

$^*$ Authors contributed equally to this paper.

## A.2 Proof of Proposition 2

We first consider the scenario when $f \in \mathcal{F}_0$. Since

$$
\begin{aligned}
f^*(t\tilde{t}) &= t\tilde{t} \cdot f\left(\frac{1}{t\tilde{t}}\right) \\
&= t\tilde{t} \cdot \left[\left(\frac{1}{t}\right)^{\gamma_0} \cdot f\left(\frac{1}{\tilde{t}}\right) + f\left(\frac{1}{t}\right)\right] \\
&= t^{1-\gamma_0} \cdot f_0^*(\tilde{t}) + f_0^*(t) \cdot \tilde{t},
\end{aligned}
$$

by letting $\gamma = 1 - \gamma_0$, we can conclude that $f_0^* \in \mathcal{F}_1$. We then consider the case when $f \in \mathcal{F}_1$. Since

$$
\begin{aligned}
f^*(t\tilde{t}) &= t\tilde{t} \cdot f\left(\frac{1}{t\tilde{t}}\right) \\
&= t\tilde{t} \cdot \left[\left(\frac{1}{t}\right)^{\gamma_1} \cdot f\left(\frac{1}{\tilde{t}}\right) + f\left(\frac{1}{t}\right) \cdot \frac{1}{\tilde{t}}\right] \\
&= t^{1-\gamma_1} \cdot f_1^*(\tilde{t}) + f_1^*(t),
\end{aligned}
$$

by letting $\gamma = 1 - \gamma_1$, we can conclude that $f_1^* \in \mathcal{F}_0$. This completes the proof. ∎

## A.3 Proof of Proposition 3

We start this proof by substituting (1), (2) and (4) into the LHS of (5)

$$
\begin{aligned}
D_{f_\lambda}(q \parallel p) &= \mathbb{E}_p[f_\lambda(q/p)] \\
&= \mathbb{E}_p[f(\lambda q/p)] - f(\lambda) \\
&= \lambda^\gamma \mathbb{E}_p[f(q/p)] + f(\lambda) \cdot \mathbb{E}_p[(p/q)^\eta] - f(\lambda).
\end{aligned}
$$

Since $f(\lambda) \cdot \mathbb{E}_p[(p/q)^\eta] = f(\lambda) \cdot \mathbb{E}_p[(p/q)^0] = f(\lambda)$ when $f \in \mathcal{F}_0$, and $f(\lambda) \cdot \mathbb{E}_p[(p/q)^\eta] = f(\lambda) \cdot \int q(x)\,dx = f(\lambda)$ when $f \in \mathcal{F}_1$, we have

$$
D_{f_\lambda}(q\|p) = \lambda^\gamma D_f(q\|p).
$$

This completes the proof. ∎

## A.4 $f$-variational bound from reverse divergence

We provide detailed steps for deriving (6), which is a preliminary step for Theorem 1 and the $f$-variational bound induced by reverse surrogate $f$-divergence. A reverse surrogate $f$-divergence can be decomposed as

$$
\begin{aligned}
D_{f_{p(\mathcal{D})^{-1}}}(q(z)\|p(z|\mathcal{D})) &= \int p(z|\mathcal{D}) \cdot f_{p(\mathcal{D})^{-1}}\left(\frac{q(z)}{p(z|\mathcal{D})}\right) dz \\
&= \int p(z|\mathcal{D}) \cdot \left[f\left(\frac{q(z) \cdot p(\mathcal{D})}{p(z,\mathcal{D})} \cdot \frac{1}{p(\mathcal{D})}\right) - f\left(\frac{1}{p(\mathcal{D})}\right)\right] dz \\
&= \frac{1}{p(\mathcal{D})} \int \frac{p(z,\mathcal{D})}{q(z)} \cdot f\left(\frac{q(z)}{p(z,\mathcal{D})}\right) \cdot q(z)\,dz - f\left(\frac{1}{p(\mathcal{D})}\right) \\
&= \frac{1}{p(\mathcal{D})} \cdot \mathbb{E}_{q(z)}\left[f^*\left(\frac{p(z,\mathcal{D})}{q(z)}\right)\right] - f\left(\frac{1}{p(\mathcal{D})}\right).
\end{aligned}
$$

## A.5 $f$-variational bound from forward divergence

As we mentioned in Section 3.1, the assumption on $p(\mathcal{D}) > 0$ or the existence of $p(\mathcal{D})^{-1}$ in (6) can be circumvented by using the $f$-VI that minimizes the forward surrogate $f$-divergence $D_{f_{p(\mathcal{D})}}(p(z|\mathcal{D})\|q(z))$. Meanwhile, in Section 3.3, the coordinate-wise update rule (16) for $f \in \mathcal{F}_0$ is also based on the $f$-variational bound induced by $D_{f_{p(\mathcal{D})}}(p(z|\mathcal{D})\|q(z))$. The $f$-variational bound

and a sandwich estimate of evidence from forward surrogate $f$-divergence are derived below. First, we notice that the forward surrogate $f$-divergence can be decomposed as follows

$$D_{f_{p(\mathcal{D})}}(p(z|\mathcal{D})\|q(z)) = \int q(z) \cdot f_{p(\mathcal{D})}\left(\frac{p(z|\mathcal{D})}{q(z)}\right) dz$$
$$= \int q(z) \cdot \left[f\left(\frac{p(z,\mathcal{D})}{q(z) \cdot p(\mathcal{D})} \cdot p(\mathcal{D})\right) - f\left(p(\mathcal{D})\right)\right] dz$$
$$= \mathbb{E}_{q(z)}\left[f\left(\frac{p(z,\mathcal{D})}{q(z)}\right)\right] - f(p(\mathcal{D})).$$

By the non-negativity of $f$-divergence [17], *i.e.* $D_{f_{p(\mathcal{D})}}(p(z|\mathcal{D})\|q(z)) \geq 0$, the $f$-variational bound $\mathcal{L}_f(q,\mathcal{D})$ from forward divergence follows

$$\mathcal{L}_f(q,\mathcal{D}) = \mathbb{E}_{q(z)}\left[f\left(\frac{p(z,\mathcal{D})}{q(z)}\right)\right] \geq f(p(\mathcal{D})), \tag{18}$$

where equality holds when $q(z) = p(z|\mathcal{D})$. Inequality (18) formulates the $f$-variational bound induced by forward divergence $D_{f_{p(\mathcal{D})}}(p(z|\mathcal{D})\|q(z))$ and supplements Theorem 1, which is based on the reverse $f$-divergence. Given convex functions $f$ and $g$ such that $f(1) = g(1) = 0$, on an interval where $f$ is non-decreasing and $g$ is non-increasing, a sandwich estimate of evidence $p(\mathcal{D})$ is given as follows

$$(g)^{-1} \circ \mathbb{E}_{q(z)}\left[g\left(\frac{p(z,\mathcal{D})}{q(z)}\right)\right] \leq p(\mathcal{D}) \leq (f)^{-1} \circ \mathbb{E}_{q(z)}\left[f\left(\frac{p(z,\mathcal{D})}{q(z)}\right)\right],$$

which supplements the sandwich estimate in Corollary 2 derived from the reverse $f$-divergence. Stochastic $f$-VI algorithms that minimize $\mathcal{L}_f(q,\mathcal{D})$ in (18) can be readily implied by imitating the steps in Section 3.2, and the optimization of $\mathcal{L}_f(q,\mathcal{D})$ in (18) also does not require $f$ and $g$ be invertible. Moreover, the statistical differences between $f$-variational bounds (8) and (18) deserve further investigations.

## A.6 Proof of Corollary 1

The proof of Corollary 1 is derived from the proof of Theorem 1 in the importance-weighted autoencoders paper [9], and we will prove Corollary 1 by utilizing the convexity of $f^*$-function and Jensen's inequality. First, we need to show that $\mathcal{L}_f^{\text{IW}}(q,\mathcal{D},L) \geq f^*(p(\mathcal{D}))$ for $L \in \mathbb{N}^*$, which is a direct result of Jensen's inequality

$$\mathcal{L}_f^{\text{IW}}(q,\mathcal{D},L) = \mathbb{E}_{z_{1:L} \sim q(z)}\left[f^*\left(\frac{1}{L}\sum_{l=1}^{L}\frac{p(z_l,\mathcal{D})}{q(z_l)}\right)\right]$$
$$\geq f^*\left(\mathbb{E}_{z_{1:L} \sim q(z)}\left[\frac{1}{L}\sum_{l=1}^{L}\frac{p(z_l,\mathcal{D})}{q(z_l)}\right]\right) = f^*(p(\mathcal{D})).$$

Next, we are to prove the statement that $\mathcal{L}_f^{\text{IW}}(q,\mathcal{D},L_1) \geq \mathcal{L}_f^{\text{IW}}(q,\mathcal{D},L_2)$ for $L_1 \leq L_2$. Let $\mathcal{I} = \{i_1, \cdots, i_{L_1}\} \subset \{1, 2, \cdots, L_2\}$ with $|\mathcal{I}| = L_1$ be a uniformly distributed subset of distinct indices from $\{1, 2, \cdots, L_2\}$. Subsequently, we have the identity $\mathbb{E}_{\mathcal{I}=\{i_1,\cdots,i_m\}}[(a_{i_1}+\cdots+a_{i_{L_1}})/L_1]$, which together with Jensen's inequality gives

$$\mathcal{L}_f^{\text{IW}}(q,\mathcal{D},L_2) = \mathbb{E}_{z_{1:L_2} \sim q(z)}\left[f^*\left(\frac{1}{L_2}\sum_{l=1}^{L_2}\frac{p(z_l,\mathcal{D})}{q(z_l)}\right)\right]$$
$$= \mathbb{E}_{z_{1:L_2} \sim q(z)}\left[f^*\left(\mathbb{E}_{I=\{i_1,\cdots,i_{L_1}\}}\left[\frac{1}{L_1}\sum_{l=1}^{L_1}\frac{p(z_l,\mathcal{D})}{q(z_l)}\right]\right)\right]$$
$$\leq \mathbb{E}_{z_{1:L_2} \sim q(z)}\left[\mathbb{E}_{I=\{i_1,\cdots,i_{L_1}\}}\left[f^*\left(\frac{1}{L_1}\sum_{l=1}^{L_1}\frac{p(z_l,\mathcal{D})}{q(z_l)}\right)\right]\right]$$
$$= \mathbb{E}_{z_{1:L_1} \sim q(z)}\left[f^*\left(\frac{1}{L_1}\sum_{l=1}^{L_1}\frac{p(z_l,\mathcal{D})}{q(z_l)}\right)\right] = \mathcal{L}_f^{\text{IW}}(q,\mathcal{D},L_1).$$

Lastly, we need to show that $f^*(p(\mathcal{D})) = \lim_{L\to\infty} \mathcal{L}_f^{\text{IW}}(q, \mathcal{D}, L)$, when $p(z, \mathcal{D})/q(z)$ is bounded. Let the random variable $R_L = \frac{1}{L} \sum_{l=1}^L p(z_l, \mathcal{D})/q(z_l)$ be bounded. By the strong law of large numbers, $R_L$ converges to $\mathbb{E}_{q(z_l)}[p(z_l, \mathcal{D})/q(z_l)] = p(\mathcal{D})$ almost surely. Therefore, $\mathcal{L}_f^{\text{IW}}(q, \mathcal{D}, L) = \mathbb{E}[f^*(R_L)]$ converges to $f^*(p(\mathcal{D}))$ *a.s.* as $L \to \infty$. This completes the proof.

# B  Examples of $f$-variational bounds

In this section, we provide some concrete examples of $f$-variational bounds by using the relationship between $f$-divergence and some specific divergences [17, 18]. Some well-known variational bounds, such as ELBO [1], RVB [2] and CUBO [3], are restored from $f$-variational bound (8)

$$\mathcal{L}_f(q, \mathcal{D}) = \mathbb{E}_{q(z)}\left[ f^*\left( \frac{p(z, \mathcal{D})}{q(z)} \right) \right] \geq f^*(p(\mathcal{D})),$$

and some new bounds that have rarely been investigated for VI are also introduced.

## B.1  $f$-variational bounds under KL divergence

The most famous variational bound induced by KL divergence is the ELBO. To restore ELBO from (8), consider a convex function $f(t) = t \cdot \log t$ with $f(1) = 0$. Hence, the dual function $f^*(t) = -\log t$ with $f^*(t) = 0$ is convex and decreasing. Substituting this $f^*$-function into (8), we have

$$\log p(\mathcal{D}) \geq \mathbb{E}_{q(z)}[\log p(z, \mathcal{D})] - \mathbb{E}_{q(z)}[\log q(z)] = \text{ELBO}, \tag{19}$$

where the RHS terms are known as the ELBO [11]. Composing both sides of (19) with an exponential function, we have a lower bound of evidence

$$p(\mathcal{D}) \geq \exp\left( \mathbb{E}_{q(z)}[\log p(z, \mathcal{D})] - \mathbb{E}_{q(z)}[\log q(z)] \right),$$

which verify the observation *o*2) and Corollary 2.

While variational upper bounds of evidence have been already discovered in Rényi's $\alpha$-VI [2] and $\chi$-VI [3], we rarely associate the variational upper bound with the classical KL-VI [1]. With the new findings in Corollary 2, we can readily define a variational upper bound subject to KL divergence. Consider the $f$-function, $f(t) = -\log t$ with $f(1) = 0$, associated with the forward KL divergence [21, 23, 25] subject to the $f$-divergence in Definition 1. The dual function then becomes $f^*(t) = t \log t$, which is decreasing on $(0, e^{-1}]$ and increasing on $(e^{-1}, \infty)$ as shown in Figure 2. Hence, substituting $f^*(t) = t \log t$ into (8), the $f$-variational bound under forward KL divergence is

$$\text{EUBO} = \mathbb{E}_{q(z)}\left[ \frac{p(z, \mathcal{D}))}{q(z)} \log\left( \frac{p(z, \mathcal{D})}{q(z)} \right) \right] \geq p(\mathcal{D}) \cdot \log p(\mathcal{D}) = f^*(p(\mathcal{D})), \tag{20}$$

where the LHS term is named as evidence upper bound (EUBO). Since $f^*(t) = t \log t$ is increasing on $(e^{-1}, \infty)$, EUBO in (20) provides an upper bound estimate of evidence when $p(\mathcal{D}) \geq e^{-1}$, which can be judged from the value of ELBO. When $p(\mathcal{D}) < e^{-1}$, one should resort to other divergences, *e.g.* $\chi$-divergence and Rényi's $\alpha$-divergence, instead of KL divergence for an upper bound of evidence. To derive an upper bound on evidence, we will only consider the occasion when $p(\mathcal{D}) \geq e^{-1}$ hereafter. According to Corollary 2, an upper bound of $p(\mathcal{D})$ can be defined by composing both sides of (20) with the inverse function of $(f^*)^{-1}(t) = t/W(t)$, which is plotted in Figure 2, and can be formulated as $(f^*)^{-1}(t) = t/W(t)$, which is well-defined on $t > 0$, and $W(t)$ is Lambert $W$ function implicitly defined by $t = W(t) \cdot \exp(W(t))$.

Hence, when $p(\mathcal{D}) \geq e^{-1}$, an upper bound induced by KL divergence can be formulated as follows

$$p(\mathcal{D}) \leq \max\{\text{EUBO}/W(\text{EUBO}), e^{-1}\}, \tag{21}$$

where EUBO is defined in (20).

## B.2  $f$-variational bounds under $\chi$-divergence

We then associate the $f$-variational bound (8) with the $\chi$-divergence, which will restore the CUBO introduced in [3]. The $\chi$-VI framework and CUBO introduced in [3] are based on minimizing the

Figure 2: $t \log t$ and its inverse function $t/W(t)$.

forward $\chi^n$-divergence $D_{\chi^n}(p\|q) = \mathbb{E}_{q(z)}[(p(z,x)/q(z))^n - 1]$ for $n \geq 1$, which is different from the reverse $\chi^n$-divergence given in Table 1. While it may be more straightforward to restore CUBO from the $f$-VI based on forward divergence introduced in Section A.5 or invoking Proposition 1 and Proposition 2 to convert the forward $\chi^n$-divergence to a reverse divergence, we will stick to the $f$-variational bound (8) and show that it is general enough to restore the CUBO with a properly chosen $f$-function. Consider an $f$-function $f(t) = t^{-1} - t$, which is convex on $t > 0$ and satisfies $f(1) = 0$. The dual function then becomes $f^*(t) = t^2 - 1$, which is increasing on $t > 0$. Substituting the dual function into (8), we have

$$\mathbb{E}_{q(z)}\left[\left(\frac{p(z,x)}{q(z)}\right)^2 - 1\right] \geq p(x)^2 - 1. \tag{22}$$

Canceling the constant terms in (22) and taking the logarithm of both sides, $\text{CUBO}_2$ follows

$$\text{CUBO}_2 = \frac{1}{2} \log \mathbb{E}_{q(z)}\left[\left(\frac{p(z,x)}{q(z)}\right)^2\right] \geq \log p(\mathcal{D}).$$

To restore the more general $\text{CUBO}_n$ for $n \in \mathbb{R} \backslash (0,1)$, we consider the $f$-function $f(t) = t^{1-n} - t$, which is convex on $t \geq 0$ and satisfies $f(1) = 0$. The corresponding dual function is $f^*(t) = t^n - 1$, which is increasing on $t > 0$ when $n \geq 1$ and decreasing on $t > 0$ when $n \leq 0$. Substituting the dual function into (8), we have

$$\mathbb{E}_{q(z)}\left[\left(\frac{p(z,x)}{q(z)}\right)^n - 1\right] \geq p(x)^n - 1. \tag{23}$$

Canceling the constant terms in (23) and taking the logarithm of both sides, $\text{CUBO}_n$ follows

$$\text{CUBO}_n = \frac{1}{n} \log \mathbb{E}_{q(z)}\left[\left(\frac{p(z,x)}{q(z)}\right)^n\right] \geq \log p(\mathcal{D}), \tag{24}$$

which gives an evidence upper bound when $n \geq 1$ and a lower bound when $n \leq 0$. When $n \in (0,1)$, a negative sign should be added such that a valid divergence is constructed [2]. When $n < 1$, $\text{CUBO}_n$ recovers the RVB in [2], which will also be briefly discussed in Section B.3. The extension to $\chi^n$-VI under reverse $\chi^n$-divergence is left to interested readers.

### B.3 $f$-variational bounds under Rényi's $\alpha$-divergence

The Rényi's $\alpha$-divergence is defined as follows

$$D_\alpha(p\|q) = \frac{1}{\alpha - 1} \log \int p(z,x)^\alpha q(z)^{1-\alpha} dz,$$

where $\alpha \in (0,1) \cup (1, +\infty)$. When $\alpha \in (-\infty, 0] \cup \{1\}$, $D_\alpha(p\|q)$ is not a valid divergence, and we will not consider this scenario, while interested readers can refer to [2] for details. Rigorously, Rényi's $\alpha$-divergence is not an $f$-divergence; however, as shown in Table 1, a one-to-one correspondence

can be established between the Rényi's $\alpha$-divergence and Hellinger $\alpha$-divergence, which is an $f$-divergence. We first show that $f$-variational bound (8) can restore the RVB when $\alpha > 1$ [2]. For $\alpha > 1$, consider an $f$-function $f(t) = t^\alpha - t$, which is convex on $t > 0$ and satisfies $f(1) = 0$. The dual function then becomes $f^*(t) = t^{1-\alpha} - 1$, which is decreasing on $t > 0$. Substituting this dual function into (8) and canceling the constant terms give

$$\mathbb{E}_{q(z)}\left[\left(\frac{p(z,\mathcal{D})}{q(z)}\right)^{1-\alpha}\right] \geq p(\mathcal{D})^{1-\alpha}. \tag{25}$$

Since $f(t) = t^\alpha - t$ is not convex when $\alpha \in (0,1)$, for this instance, we then consider the function $f(t) = -t^\alpha + t$, which is convex on $t > 0$ and satisfies $f(1) = 0$. The dual function is $f^*(t) = -t^{1-\alpha} + 1$, which is decreasing on $t > 0$ and $\alpha \in (0,1)$. Substituting this dual function into (8) and canceling the constant terms give

$$\mathbb{E}_{q(z)}\left[\left(\frac{p(z,\mathcal{D})}{q(z)}\right)^{1-\alpha}\right] \leq p(\mathcal{D})^{1-\alpha}. \tag{26}$$

Taking the logarithm on both sides of (25) and (26), and dividing both sides of the results by $1 - \alpha$, we have

$$\text{RVB} = \frac{1}{1-\alpha} \log \mathbb{E}_{q(z)}\left[\left(\frac{p(z,\mathcal{D})}{q(z)}\right)^{1-\alpha}\right] \leq \log p(\mathcal{D}), \tag{27}$$

which is identical to the RVB $\mathcal{L}_{\alpha+}(q;\mathcal{D})$ defined in [2].

### B.4  $f$-variational bounds under total variation distance

The total variation distance is induced by the $f$-function $f(t) = |t-1|$ with dual function $f^*(t) = |t-1| = f(t)$. This $f$-function poses stark differences than the previous examples: i) $f$- and $f^*$-functions are not smooth at $t = 1$, ii) $f$- and $f^*$-functions are not monotonic on $t > 0$, and iii) the dual function $f^*(t) = |t-1|$ is not invertible. Nonetheless, since the dual function $f^*(t) = |t-1|$ is decreasing on $t \in [0,1)$ and increasing on $t \in (1,\infty)$, the $f$-variational bounds subject to total variation can still provide a valid upper/lower bound of evidence on each monotonic interval. Substituting $f^*(t) = |t-1| = f(t)$ into (8), we have

$$\mathbb{E}_{q(z)}\left[\left|\frac{p(z,\mathcal{D})}{q(z)} - 1\right|\right] \geq |p(\mathcal{D}) - 1|. \tag{28}$$

When $p(\mathcal{D}) \in [0,1)$, inequality (28) gives a lower bound of evidence

$$p(\mathcal{D}) \geq 1 - \mathbb{E}_{q(z)}\left[\left|\frac{p(z,\mathcal{D})}{q(z)} - 1\right|\right]. \tag{29}$$

When $p(\mathcal{D}) \geq 1$, inequality (28) gives an upper bound of evidence

$$p(\mathcal{D}) \leq 1 + \mathbb{E}_{q(z)}\left[\left|\frac{p(z,\mathcal{D})}{q(z)} - 1\right|\right]. \tag{30}$$

Combining (29) and (30), the $f$-variational bounds induced by the total variation distance are given as follows

$$\max\left\{0, 1 - \mathbb{E}_{q(z)}\left[\left|\frac{p(z,\mathcal{D})}{q(z)} - 1\right|\right]\right\} \leq p(\mathcal{D}) \leq 1 + \mathbb{E}_{q(z)}\left[\left|\frac{p(z,\mathcal{D})}{q(z)} - 1\right|\right]. \tag{31}$$

A vanilla example demonstrating the $f$-variational bounds associated with total variation distance is provided in Figure 3 of Section E.1.

## C  Examples of stochastic $f$-variational inference

This section provides supplementary interpretations for Section 3.2 with i) steps for deriving the score function gradient in (10), ii) concrete examples of the score function, reparameterization, and IW-reparameterization gradients under KL, $\chi$-, and Rényi's $\alpha$-divergences, and iii) a reference algorithm

for black box (stochastic) $f$-VI. First, we derive the score function gradient (10) for optimizing the parameters $\theta$ in recognition model $q_\theta(z)$. Computing the gradient of $f$-variational bound $\mathcal{L}_f(q_\theta, \mathcal{D})$ in (8) w.r.t. parameters $\theta$, we have

$$
\begin{aligned}
\nabla_\theta \mathcal{L}_f(q_\theta, \mathcal{D}) = \nabla_\theta \mathbb{E}_{q_\theta(z)} \left[ f^* \left( \frac{p(z, \mathcal{D})}{q_\theta(z)} \right) \right] &= \int p(z, \mathcal{D}) \cdot \nabla_\theta f \left( \frac{q_\theta(z)}{p(z, \mathcal{D})} \right) dz \\
&= \int q_\theta(z) \cdot f' \left( \frac{q_\theta(z)}{p(z, \mathcal{D})} \right) \cdot \frac{\nabla_\theta q_\theta(z)}{q_\theta(z)} \, dz \\
&= \mathbb{E}_{q_\theta(z)} \left[ f' \left( \frac{q_\theta(z)}{p(z, \mathcal{D})} \right) \cdot \nabla_\theta \log q_\theta(z) \right],
\end{aligned}
$$

where $f'(t)$ denotes $\partial f(t)/\partial t$. An unbiased MC estimator for this score gradient function is given in (11).

### C.1  Gradient estimators under KL divergence

We first provide the gradient estimators for stochastic $f$-VI subject to KL divergence. For the ELBO originated from reverse KL divergence, we choose the $f$-function $f(t) = t \log t$, which gives the dual function $f^*(t) = -\log t$ and derivative $f'(t) = 1 + \log t$. Substituting $f'(t) = 1 + \log t$ into (11) and multiplying the result by $-1$[5], we have a score function gradient estimator of ELBO

$$
\nabla_\theta \hat{\mathcal{L}}_f(q_\theta, \mathcal{D}) = \frac{1}{K} \sum_{k=1}^{K} \log \frac{p(z_k, \mathcal{D})}{q_\theta(z_k)} \cdot \nabla_\theta \log q_\theta(z_k), \tag{32}
$$

where $z_k \sim q_\theta(z)$. The score function gradient estimator (32) for ELBO restores the result in [26]. Given a noise variable $\varepsilon \sim p(\varepsilon)$ and a mapping $g_\theta(\cdot)$ such that $z = g_\theta(\varepsilon)$, and substituting $f^*(t) = -\log t$ into (13) and multiplying the result by $-1$, we have a reparameterization gradient estimator of ELBO

$$
\nabla_\theta \hat{\mathcal{L}}_f^{\text{rep}}(q_\theta, \mathcal{D}) = \frac{1}{K} \sum_{k=1}^{K} \nabla_\theta \log \frac{p(g_\theta(\varepsilon_k), \mathcal{D})}{q(g_\theta(\varepsilon_k))}, \tag{33}
$$

where $\varepsilon_k \sim p(\varepsilon)$. The reparameterization gradient (33) restores the gradient of standard VAE in [7]. Substituting $f^*(t) = -\log t$ into (14) and drawing the two-dimensional noise samples $\{\varepsilon_{k,1:L}\}_{k=1}^{K}$ from $p(\varepsilon)$, we have an IW-reparameterization gradient of ELBO

$$
\nabla_\theta \hat{\mathcal{L}}_f^{\text{IW, rep}}(q_\theta, \mathcal{D}, L) = \frac{1}{K} \sum_{k=1}^{K} \nabla_\theta \log \left( \frac{1}{L} \sum_{l=1}^{L} \frac{p(g_\theta(\varepsilon_{k,l}), \mathcal{D})}{q_\theta(g_\theta(\varepsilon_{k,l}))} \right),
$$

which restores the gradient of IW-VAE in [9]. In practice, the (IW-)reparameterization gradients can be computed by invoking the backpropagation functions in machine learning libraries or other automatic differentiation tools.

We then give the gradients for optimizing the EUBO defined in (20), which has rarely been reported before. For EUBO, we consider the $f$-function $f(t) = -\log t$, which gives the dual function $f^*(t) = t \log t$ and derivative $f'(t) = -1/t$. Hence, substituting $f'(t) = -1/t$ into (11), we have a score function gradient estimator of EUBO

$$
\nabla_\theta \hat{\mathcal{L}}_f(q_\theta, \mathcal{D}) = -\frac{1}{K} \sum_{k=1}^{K} \frac{p(z_k, \mathcal{D})}{q_\theta(z_k)} \cdot \nabla_\theta \log q_\theta(z_k),
$$

where $z_k \sim q_\theta(z)$. The reparameterization gradient estimator of EUBO can be obtained by substituting the dual function $f^*(t) = t \log t$ into (13), which gives

$$
\nabla_\theta \hat{\mathcal{L}}_f^{\text{rep}}(q_\theta, \mathcal{D}) = \frac{1}{K} \sum_{k=1}^{K} \nabla_\theta \left( \frac{p(g_\theta(\varepsilon_k), \mathcal{D}))}{q_\theta(g_\theta(\varepsilon_k))} \cdot \log \frac{p(g_\theta(\varepsilon_k), \mathcal{D})}{q_\theta(g_\theta(\varepsilon_k))} \right),
$$

where noise samples $\varepsilon_k \sim p(\varepsilon)$. The IW-reparameterization gradient estimator of EUBO is obtained by substituting the dual function $f^*(t) = t \log t$ into (14), which gives

$$\nabla_\theta \hat{\mathcal{L}}_f^{\text{IW, rep}}(q_\theta, \mathcal{D}, L) = \frac{1}{K} \sum_{k=1}^K \nabla_\theta \left( \frac{1}{L} \sum_{l=1}^L \frac{p(g_\theta(\varepsilon_{k,l}), \mathcal{D})}{q_\theta(g_\theta(\varepsilon_{k,l}))} \cdot \log \left( \frac{1}{L} \sum_{l=1}^L \frac{p(g_\theta(\varepsilon_{k,l}), \mathcal{D})}{q_\theta(g_\theta(\varepsilon_{k,l}))} \right) \right),$$

where the noise samples $\{\varepsilon_{k,1:L}\}_{k=1}^K \sim p(\varepsilon)$.

## C.2   Gradient estimators under $\chi$-divergence

We then implement the gradient estimators of $f$-VI to $\chi$-divergence. For conciseness, we only consider the gradient of objective function $\exp(n \cdot \text{CUBO}_n)$, which has unbiased estimators, while the estimators of $\text{CUBO}_n$ in (24) are biased but more stable in numerical computation. Similar to Section B.2, we choose the $f$-function $f(t) = t^{1-n} - t$, which implies the dual function $f^*(t) = t^n - 1$ and the derivative $f'(t) = (1-n)t^{-n} - 1$. Hence, substituting $f'(t) = (1-n)t^{-n} - 1$ into (11), we have a score function gradient estimator for $\chi$-VI

$$\nabla_\theta \hat{\mathcal{L}}_f(q_\theta, \mathcal{D}) = \frac{1-n}{K} \sum_{k=1}^K \left[ \left( \frac{p(z_k, \mathcal{D})}{q_\theta(z_k)} \right)^n \nabla_\theta \log q_\theta(z_k) \right]$$

where $z_k \sim q_\theta(z)$. Given a noise variable $\varepsilon \sim p(\varepsilon)$ and a mapping $g_\theta(\cdot)$ such that $z = g_\theta(\varepsilon)$, the reparameterization gradient estimator is obtained by substituting $f^*(t) = t^n - 1$ into (13)

$$\nabla_\theta \hat{\mathcal{L}}_f^{\text{rep}}(q_\theta, \mathcal{D}) = \frac{1}{K} \sum_{k=1}^K \nabla_\theta \left( \frac{p(g_\theta(\varepsilon_k), x)}{q_\theta(g_\theta(\varepsilon_k))} \right)^n = \frac{n}{K} \sum_{k=1}^K \left( \frac{p(g_\theta(\varepsilon_k), x)}{q_\theta(g_\theta(\varepsilon_k))} \right)^n \nabla_\theta \log \frac{p(g_\theta(\varepsilon_k), x)}{q_\theta(g_\theta(\varepsilon_k))},$$

where noise samples $\varepsilon_k \sim p(\varepsilon)$. While the preceding two gradient estimators recover the result in [3], we supplement $\chi$-VI with an IW-reparameterization gradient estimator, which is obtained by substituting $f^*(t) = t^n - 1$ into (14)

$$\nabla_\theta \hat{\mathcal{L}}_f^{\text{IW, rep}}(q_\theta, \mathcal{D}, L) = \frac{n}{K} \sum_{k=1}^K \left( \frac{1}{L} \sum_{l=1}^L \frac{p(g_\theta(\varepsilon_{k,l}), \mathcal{D})}{q_\theta(g_\theta(\varepsilon_{k,l}))} \right)^n \nabla_\theta \log \left( \frac{1}{L} \sum_{l=1}^L \frac{p(g_\theta(\varepsilon_{k,l}), \mathcal{D})}{q_\theta(g_\theta(\varepsilon_{k,l}))} \right),$$

where the noise samples $\{\varepsilon_{k,1:L}\}_{k=1}^K \sim p(\varepsilon)$.

## C.3   Gradient estimators under Rényi's $\alpha$-divergence

Our last example implements the $f$-VI gradient estimators to Rényi's $\alpha$-divergences and supplements Rényi's $\alpha$-VI [2] with a set of unbiased gradient estimators. Similar to the gradients of $\chi$-VI introduced in Section C.2, this section considers the gradient estimators of objective function $\exp\{(1-\alpha) \cdot \text{RVB}\}$, where RVB is defined in (27). The choices of $f$-functions are i) $f(t) = t^\alpha - t$, $f^*(t) = t^{1-\alpha} - 1$ and $f'(t) = \alpha t^{\alpha-1} - 1$ for $\alpha \in (0, 1)$, and ii) $f(t) = -t^\alpha + t$, $f^*(t) = -t^{1-\alpha} + 1$, and $f'(t) = -\alpha t^{\alpha-1} + 1$ for $\alpha \in (1, +\infty)$. Consequently, the score gradient estimator is

$$\nabla_\theta \hat{\mathcal{L}}_f(q_\theta, \mathcal{D}) = \frac{\alpha}{K} \sum_{k=1}^K \left( \frac{q_\theta(z_k)}{p(z_k, \mathcal{D})} \right)^{\alpha-1} \nabla_\theta \log q_\theta(z_k),$$

where $z_k \sim q_\theta(z)$. Given a noise variable $\varepsilon \sim p(\varepsilon)$ and a mapping $g_\theta(\cdot)$ such that $z = g_\theta(\varepsilon)$, the reparameterization gradient estimator under Rényi's $\alpha$-divergence is

$$\nabla_\theta \hat{\mathcal{L}}_f^{\text{rep}}(q_\theta, \mathcal{D}) = \frac{1-\alpha}{K} \sum_{k=1}^K \left( \frac{p(g_\theta(\varepsilon_k), \mathcal{D})}{q_\theta(g_\theta(\varepsilon_k))} \right)^{1-\alpha} \nabla_\theta \log \frac{p(g_\theta(\varepsilon_k), x)}{q_\theta(g_\theta(\varepsilon_k))},$$

where noise samples $\varepsilon_k \sim p(\varepsilon)$. The IW-reparameterization gradient estimator then becomes

$$\nabla_\theta \hat{\mathcal{L}}_f^{\text{rep}}(q_\theta, \mathcal{D}, L) = \frac{1-\alpha}{K} \sum_{k=1}^K \left( \frac{1}{L} \sum_{l=1}^L \frac{p(g_\theta(\varepsilon_{k,l}), \mathcal{D})}{q_\theta(g_\theta(\varepsilon_{k,l}))} \right)^{1-\alpha} \nabla_\theta \log \left( \frac{1}{L} \sum_{l=1}^L \frac{p(g_\theta(\varepsilon_{k,l}), \mathcal{D})}{q_\theta(g_\theta(\varepsilon_{k,l}))} \right),$$

where the noise samples $\{\varepsilon_{k,1:L}\}_{k=1}^K \sim p(\varepsilon)$.

## C.4 Stochastic $f$-variational inference algorithm

The following table provides a reference algorithm to implement stochastic $f$-VI.

---

**Algorithm 1:** Stochastic $f$-VI

---

**Input:** Dataset $\mathcal{D} = \{x_n\}_{n=1}^N$, model $p(z, x)$, variational family $q_\theta(z)$, and $f$-function.
**Initialize:** Recognition parameters $\theta_0$.
**while** $\theta$ *has not converged* **do**
  Randomly draw i) a minibatch $\mathcal{D}_M$ from full dataset $\mathcal{D}$ and ii) nosise samples $\{\varepsilon_k\}_{k=1}^K$ or
    $\{\varepsilon_{k,1:L}\}_{k=1}^K$ from noise distribution $p(\boldsymbol{\varepsilon})$;
  Approximate full likelihood $p(g_{\theta_t}(\varepsilon_k), D)$, recognition distribution $q_{\theta_t}(g_{\theta_t}(\varepsilon_k))$, and prior
    distribution $p(g_{\theta_t}(\varepsilon_k))$;
  Compute the gradient of $f$-variational bound from (11), (13) or (14);
  Update parameters $\theta_{t+1}$ from $\theta_t$ and the gradient.
**end**
**Return:** Recognition distribution $q_\theta(z)$.

---

# D   Mean-field $f$-variational inference

This section supplements the mean-field $f$-VI by providing i) steps for deriving the coordinate-wise update rules (15) and (16), ii) an example of mean-field $f$-VI subject to KL divergence, and iii) a reference mean-field $f$-VI algorithm. The mean-field $f$-VI is developed on the basis of mean-field assumption $q(z) = \prod_{j=1}^J q_j(z_j)$ and $f$-function's homogeneity decomposition $f \in \mathcal{F}_{\{0,1\}}$.

## D.1   Deviation of update rules

We first show the detailed steps for deriving the coordinate-wise update rules (15) and (16) in mean-field $f$-VI. For conciseness, we define $p = p(z, \mathcal{D})$ and $q = q(z) = q_j(z_j) \cdot \prod_{\ell \neq j} q_\ell(z_\ell) = q_j \cdot q_{-j}$. The update rules are then derived by singling out the term $q_j$ from $f$-variational bounds (8) or (18) while fixing all the other terms that consist of $q_{-j}$. For $f$-divergences with $f \in \mathcal{F}_1$ or $f^* \in \mathcal{F}_0$, such as KL divergence, we have $f(t\tilde{t}) = t^\gamma f(\tilde{t}) + f(t)\tilde{t}$ and $f^*(t\tilde{t}) = t^{1-\gamma} f^*(\tilde{t}) + f^*(t)$. Hence, the $f$-variational bound (8) can be reformulated as

$$
\begin{aligned}
\mathcal{L}_f(q_j, q_{-j}, \mathcal{D}) = \mathbb{E}_q \left[ \frac{p}{q} \cdot f\left( \frac{q}{p} \right) \right] &= \mathbb{E}_q \left[ f^* \left( \frac{1}{q_j} \cdot \frac{p}{q_{-j}} \right) \right] \\
&= \mathbb{E}_q \left[ q_j^{\gamma-1} \cdot f^* \left( \frac{p}{q_{-j}} \right) \right] + \mathbb{E}_q \left[ f^* \left( \frac{1}{q_j} \right) \right] \\
&= \mathbb{E}_{q_j} \left[ q_j^{\gamma-1} \cdot \mathbb{E}_{q_{-j}} \left[ f^* \left( \frac{p}{q_{-j}} \right) \right] \right] + \mathbb{E}_{q_j} \left[ f^* \left( \frac{1}{q_j} \right) \right] \\
&= \mathbb{E}_{q_j} \left[ q_j^{\gamma-1} \cdot f^* \circ f^{*-1} \left( \mathbb{E}_{q_{-j}} \left[ f^* \left( \frac{p}{q_{-j}} \right) \right] \right) \right] + \frac{1}{q_j} \cdot f(q_j) \\
&= \mathbb{E}_{q_j} \left[ \frac{m_j}{q_j} \cdot \left( q_j^\gamma \cdot f\left( \frac{1}{m_j} \right) + f(q_j) \cdot \frac{1}{m_j} \right) \right] \\
&= \mathbb{E}_{q_j} \left[ \frac{m_j}{q_j} \cdot f\left( \frac{q_j}{m_j} \right) \right] = \mathbb{E}_{q_j} \left[ f^* \left( \frac{m_j}{q_j} \right) \right],
\end{aligned}
$$

where $m_j = f^{*-1}(\mathbb{E}_{q_{-j}}[f^*(p/q_{-j})])$ can be regarded as an unnormalized probability distribution. After normalizing $m_j$ into a probability distribution $\tilde{m}_j$ with normalization constant $c > 0$, the $f$-variational bound then becomes $\mathcal{L}_f(q_j, q_{-j}, \mathcal{D}) = c \cdot D_{f^*}(\tilde{m}_j \| q_j)$, which attains its minimum at $\tilde{m}_j = q_j$. Therefore, to minimize the $f$-variational bound when $f \in \mathcal{F}_1$, the marginal distribution $q_j$ should be updated in accordance with (15):

$$
q_j \propto m_j = f^{*-1} \left( \mathbb{E}_{q_{-j}} \left[ f^* \left( \frac{p(z, \mathcal{D})}{q_{-j}(z_{-j})} \right) \right] \right).
$$

For $f$-divergences with $f \in \mathcal{F}_0$ or $f^* \in \mathcal{F}_1$, such as $\chi$- and Rényi's $\alpha$-divergences, we have identity $f(t\tilde{t}) = t^\gamma f(\tilde{t}) + f(t)$, and the coordinate-wise update rule for these divergences is derived by singling out $q_j$ from the variational bound of forward $f$-divergence VI (18) introduced in Section A.5. Hence, the $f$-variational bound (18) can be reformulated as

$$
\begin{aligned}
\mathcal{L}_f(q_j, q_{-j}, \mathcal{D}) &= \mathbb{E}_q\left[f\left(\frac{p}{q}\right)\right] = \mathbb{E}_q\left[f\left(\frac{1}{q}\cdot p\right)\right] \\
&= \mathbb{E}_q\left[\left(\frac{1}{q_j\cdot q_{-j}}\right)^\gamma \cdot f(p) + f\left(\frac{1}{q_j\cdot q_{-j}}\right)\right] \\
&= \mathbb{E}_q\left[\left(\frac{1}{q_j}\right)^\gamma\left(\frac{1}{q_{-j}}\right)^\gamma f(p) + \left(\frac{1}{q_j}\right)^\gamma f\left(\frac{1}{q_{-j}}\right) + f\left(\frac{1}{q_j}\right)\right] \\
&= \mathbb{E}_{q_j}\left[\left(\frac{1}{q_j}\right)^\gamma \cdot \mathbb{E}_{q_{-j}}\left[\left(\frac{1}{q_{-j}}\right)^\gamma f(p) + f\left(\frac{1}{q_{-j}}\right)\right] + f\left(\frac{1}{q_j}\right)\right] \\
&= \mathbb{E}_{q_j}\left[\left(\frac{1}{q_j}\right)^\gamma \cdot f\circ f^{-1}\left(\mathbb{E}_{q_{-j}}\left[f\left(\frac{p}{q_{-j}}\right)\right]\right) + f\left(\frac{1}{q_j}\right)\right] \\
&= \mathbb{E}_{q_j}\left[\left(\frac{1}{q_j}\right)^\gamma f(m_j) + f\left(\frac{1}{q_j}\right)\right] \\
&= \mathbb{E}_{q_j}\left[f\left(\frac{m_j}{q_j}\right)\right],
\end{aligned}
$$

where $m_j = f^{-1}(\mathbb{E}_{q_{-j}}[f(p/q_{-j})])$ can be regarded as an unnormalized probability distribution. After scaling and normalizing $m_j$ into a probability distribution $\tilde{m}_j$ with normalization constant $c > 0$, we have $\mathcal{L}_f(q_j, q_{-j}, \mathcal{D}) = c \cdot D_f(\tilde{m}_j \| q_j)$, which attains its minimum at $\tilde{m}_j = q_j$. Therefore, to minimize the $f$-variational bound when $f \in \mathcal{F}_0$, the marginal distribution $q_j$ should be updated with (16):

$$
q_j \propto m_j = f^{-1}\left(\mathbb{E}_{q_{-j}}\left[f\left(\frac{p(z,\mathcal{D})}{q_{-j}(z_{-j})}\right)\right]\right). \tag{34}
$$

## D.2 Mean-field $f$-variational inference under KL divergence

For mean-field $f$-VI, we only show an example associated with KL divergence. For KL divergence, consider the $f$-function $f(t) = t\log t \in \mathcal{F}_1$ with $f^*(t) = -\log t$ and $f^{*-1}(t) = \exp(-t)$. Hence, the coordinate-wise update rule (15) takes the form

$$
q_j^* \propto \exp\left(\mathbb{E}_{q_{-j}}[\log p(z,\mathcal{D})] - \mathbb{E}_{q_{-j}}[\log q_{-j}]\right) \propto \exp\left(\mathbb{E}_{q_{-j}}[\log p(z,\mathcal{D})]\right),
$$

which is in accordance with the update rule of CAVI algorithm [15]. Demonstrations and experimental results of this update rule can be easily found in the early developments of KL-VI [11, 15, 23]. An analytic update rule requires conditionally conjugate models, while some recent advances tried to extend mean-field VI to non-conjugate models [4, 5]. Mean-field $f$-VI subject to other divergences are left to the interested readers to explore.

## D.3 Mean-field $f$-variational inference algorithm

A reference algorithm to implement mean-field $f$-VI is given in the following table.

---
**Algorithm 2:** Mean-field $f$-VI

---
**Input:** Dataset $\mathcal{D} = \{x_n\}_{n=1}^N$, mean-field variational family $q(z,\theta) = \prod_{j=1}^J q_j(z_j, \theta_j)$, model
    $p(z,x)$, $f$-function $f(\cdot)$, and $f$-variational bound $\mathcal{L}_f(q_\theta, \mathcal{D})$.
**Initialize:** Variational parameters $\theta$ in recognition model $q(z,\theta)$.
**while** $\mathcal{L}_f(q_\theta, \mathcal{D})$ *has not converged* **do**
    Update parameters $\theta_j$ in $q_j(z_j, \theta_j)$ for $j \in \{1, \cdots, J\}$ with update rule (15) or (16);
    Compute $f$-variational bound $\mathcal{L}_f(q_\theta, \mathcal{D})$.
**end**
**Return:** Recognition distribution $q(z,\theta)$.

---

# E  Experiments

Detailed descriptions on the experimental settings and the supplementary empirical results are provided in this section.

## E.1  Synthetic example

For the synthetic example in the original paper, we consider a batch of i.i.d. datapoints generated by the latent variable model $x = \sin(z) + \mathcal{N}(0, 0.01)$, $z \sim \text{UNIF}(0, \pi)$. To estimate the true evidence $p(\mathcal{D})$ and $f$-variational bounds, we posit a prior distribution $p(z) = \text{UNIF}(0, \pi)$, a likelihood distribution $p(z|x) = \mathcal{N}(\sin(z), 0.01)$, and an approximate model $q_\theta(z) = \text{UNIF}(\frac{1-\theta}{2}\pi, \frac{\theta+1}{2}\pi)$, which is a uniform distribution centered at $z = \pi/2$ with width $\theta\pi$. The true evidence $p(\mathcal{D})$ is approximated by a naive MC estimator $\hat{p}(x) = \sum_{k=1}^{K} p(x|z_k)$ with $K = 5 \times 10^5$, and all the other (importance-weighted) $f$-variational bounds in Figure 1 and Figure 3 are estimated by their corresponding MC estimators with $L = 8$ and $K = 5 \times 10^4$. Fixing $\theta = 1.1$, we approximate the importance-weighted $f$-variational bound subject to total variation distance (IW-TVB) in Figure 3, which verifies (31) and Corollary 2.

Figure 3: Evdience and IW-TVB.

However, it is still worth noting that numerical issues and biased estimators can contaminate the empirical results or cause the violations of theory, despite the fact that the importance-weighted technique can attenuate these flaws by improving the tightness of bounds and their estimators. The estimation of IW-RVB in Figure 1 and IW-TVB on $x \in [-0.5, 0]$ in Figure 3 are some examples. More discussions and examples on these problems can be found in [2, 10].

## E.2  Bayesian neural network

Our Bayesian regression framework is developed on the basis of [2]. The regression model is a single ReLU layer with 50 hidden units for small datasets and 100 hidden units for large datasets (Protein). The likelihood function is selected as $p(y|x, z) = \mathcal{N}(y; F_z(x), \sigma^2)$, where $\sigma$ is a hyper-parameter and $F_z(x)$ is the prediction or output of the neural network with weights $z$. We posit a standard normal prior $z \sim \mathcal{N}(0, I)$ for network weights and a Gaussian approximation $q(z) = \mathcal{N}(\mu_\theta, \text{diag}(\sigma_\theta^2))$ to the true posterior, where the variational parameters $\mu_\theta$ and $\sigma_\theta$ are to be optimized. Importance-weighted $f$-variational bounds and their gradients are approximated by MC estimators with $L = 5$, $K = 50$ for small datsets and $K = 10$ for large datasets. Twelve datasets from the UCI Machine Learning Repository [36] are employed, in which six datasets (Boston, CCPP, Concrete, Protein, Wine and Yacht) are the benchmarks previously tested in [2, 10, 20], while the other six datasets[6] are new benchmarks for VI testing. Each dataset is randomly split into $90\%/10\%$ for training and testing. The test RMSE and test negative log-likelihood reported in Table 2 are collected from 20 trials with 500 training epochs in each trial for small datasets and 5 trials with 200 training epochs in each trial

for large datasets. For dataset Building, we predict the sale prices and scale the test RMSE by 0.01 for uniform representation. For dataset Stock, we only use the 5-year data to predict the annual return, and the test RMSEs are scaled by 100.

Six $f$-VIs, including three well-established $f$-VIs (KL-VI, Rényi's $\alpha$-VI with $\alpha = 3$, and $\chi$-VI with $n = 2$) and three new $f$-VIs (VIs subject to total variation distance and two custom $f$-divergences), are tested and compared in this Bayesian regression example and the following $f$-VAE example. The total variation bound is defined as TVB $= \mathbb{E}_{q(z)}\left[|p(z, \mathcal{D})/q(z) - 1|\right]$, and since $p(z, \mathcal{D})/q(z) \in (0, 1)$ always holds in Section 4.2 and Section 4.3, we optimize the objective function $\log(\text{TVB} - 1)$ for numerical stability. Meanwhile, we also consider i) a custom $f$-VI induced by the dual function $f_{c1}^*(t) = \tilde{f}^*(t) - \tilde{f}^*(1)$, where $\tilde{f}^*(t) = -1/6 \cdot (\log t + t_0)^3 - 1/2 \cdot (\log t + t_0)^2 - (\log t + t_0) - 1$, $t = p(z, \mathcal{D})/q(z)$, and $t_0 \in \mathbb{R}$ is a parameter to be optimized, and ii) a custom $f$-VI induced by the dual function $f_{c2}^*(t) = \log^2 t + \log t$, which is convex on $t \in (0, 1)$ and can be modified to be a valid $f$-function by reassigning the mapping on $t \in [1, +\infty)$. More feasible $f^*$-functions can be generated from the known $f^*$-functions via the operations that preserve convexity, *e.g.* non-negative weighted sums. While the $f$-VI framework applies to arbitrary valid $f$-functions in theory, the empirical implementations require the $f$-functions and the corresponding estimators to have good numerical properties such that the optimization algorithms can converge. To meet this requirement, we sometimes have to compromise the unbiasedness of estimators, for example, while the CUBO ($n = 2$) employed for regression in Table 2 should be an upper bound of evidence in theory, the empirical CUBO approximated by a biased estimator in [2] behaves like a lower bound in the training processes, despite the augmentation of importance-weighted technique.

Table 2a: Average test error.

| Dataset | Test RMSE (lower is better) | | | | | |
|---------|--------|--------|--------|--------|--------|--------|
| | KL-VI | $\chi$-VI | $\alpha$-VI | TV-VI | $f_{c1}$-VI | $f_{c2}$-VI |
| Airfoil | **2.16±.07** | 2.36±.14 | 2.30±.08 | 2.47±.15 | 2.34±.09 | 2.16±.09 |
| Aquatic | **1.12±.06** | 1.20±.06 | 1.14±.07 | 1.23±.10 | 1.14±.06 | 1.14±.06 |
| Boston | **2.76±.36** | 2.99±.37 | 2.86±.36 | 2.96±.36 | 2.87±.36 | 2.89±.38 |
| Building | 1.38±.12 | 2.82±.51 | 1.83±.22 | 2.57±.59 | 1.80±.21 | **1.36±.15** |
| CCPP | **4.05±.09** | 4.14±.11 | 4.06±.08 | 4.19±.12 | 4.33±.12 | 4.33±.12 |
| Concrete | 5.40±.24 | **3.32±.34** | 5.32±.27 | 5.27 ±.24 | 5.26±.21 | 5.32±.24 |
| Fish Toxicity | .885±.037 | .905±.043 | .891±.037 | .878±.044 | .883±.034 | **.862±.040** |
| Protein | 1.93±.19 | 2.45±.42 | **1.87±.17** | 2.91±.89 | 1.97±.21 | 1.97±.20 |
| Real Estate | 7.48±1.41 | 7.51±1.44 | **7.46±1.42** | 8.02±1.58 | 7.52±1.40 | 7.99±1.55 |
| Stock | 3.85±1.12 | 3.90±1.09 | 3.88±1.13 | 4.33±.43 | **3.82±1.11** | 4.18±.42 |
| Wine | .642±.018 | .640±.021 | .638±.018 | .645±.014 | .643±.019 | **.637±.016** |
| Yacht | **0.78±.12** | 1.18±.18 | 0.99±.12 | 1.03±.14 | 1.00±.18 | 0.82±.16 |

Table 2b: Average negative log-likelihood.

| Dataset | Test negative log-likelihood (lower is better) | | | | | |
|---------|--------|--------|--------|--------|--------|--------|
| | KL-VI | $\chi$-VI | $\alpha$-VI | TV-VI | $f_{c1}$-VI | $f_{c2}$-VI |
| Airfoil | **2.17±.03** | 2.27±.03 | 2.26±.02 | 2.28±.04 | 2.29±.02 | 2.18±.03 |
| Aquatic | **1.54±.04** | 1.60±.08 | 1.54±.07 | 1.56±.07 | 1.54±.06 | 1.55±.04 |
| Boston | 2.49±.08 | 2.54±.18 | **2.48±.13** | 2.51±.18 | 2.49±.13 | 2.51±.10 |
| Building | 6.62±.02 | 6.94±.13 | 6.79±.03 | 6.88±.08 | 6.74±.04 | **6.55±.02** |
| CCPP | **2.82±.02** | 2.84±.03 | 2.82±.02 | 2.83±.02 | 2.95±.01 | 2.91±.01 |
| Concrete | 3.10±.04 | **2.61±.18** | 3.09±.04 | 3.10±.05 | 3.09±.03 | 3.10±.04 |
| Fish Toxicity | 1.28±.04 | 1.27±.04 | 1.29±.04 | **1.26±.05** | 1.29±.03 | 1.26±.03 |
| Protein | **2.00±.07** | 2.01±.08 | 2.04±.08 | 2.04±.11 | 2.21±.04 | 2.11±.05 |
| Real Estate | 3.60±.30 | 3.70±.45 | **3.59±.32** | 3.86±.52 | 3.62±.33 | 3.74±.37 |
| Stock | -1.09±.04 | -1.09±.04 | -1.09±.04 | -1.73±.15 | -1.09±.04 | **-1.84±.12** |
| Wine | .966±.027 | .965±.028 | .964±.025 | .969±.023 | .975±.027 | **.959±.023** |
| Yacht | **1.70±.02** | 1.79±.03 | 1.82±.01 | 1.78±.02 | 2.05±.01 | 1.86±.02 |

### E.3 Bayesian variational autoencoder

Our Bayesian VAE example is built on the basis of [41]. The encoder network downsamples from a $28 \times 28$ or $28 \times 20$ image to a 20-dimensional latent space and sequentially consists of i) a $3 \times 3$

2-D convolution layer with stride 2, ii) a ReLU layer, iii) a $3 \times 3$ 2-D convolution layer with stride 2, iv) a ReLU layer, and v) a fully connected layer. The decoder network scales up the 20-dimensional encoding back into a $28 \times 28$ or $28 \times 20$ image and sequentially consists of i) a $7 \times 7$ or $7 \times 5$ transposed 2-D convolution layer with stride $[7, 7]$ or $[7, 5]$, ii) a ReLU layer, iii) a $3 \times 3$ transposed 2-D convolution layer with stride 2, iv) a ReLU layer, v) a $3 \times 3$ transposed 2-D convolution layer with stride 2, vi) a ReLU layer, and vii) a $3 \times 3$ transposed 2-D convolution layer. The sizes of training/testing datasets are respectively, 7803/868, 1768/197, 60000/10000, and 24345/8070 for Caltech 101 Silhouettes, Frey Face, MNIST, and Omniglot, and the mini-batch sizes are respectively 64, 32, 512, and 256. The loss functions or the importance-weighted $f$-variational bounds are approximated by single-sample MC estimators with $K = 1$ and $L = 3$. After 20 trials with 200 training epochs in each trial, the average test reconstruction errors (lower is better) measured by cross-entropy are given in Table 3. Some reconstructed and generated images from $f$-VAEs are presented in Figure 4 to Figure 8. While one can improve the quality of these images and reduce the average reconstruction errors in Table 3 by adopting more complex encoder and decoder networks, in this experiment, we are more interested in the relative performance of different $f$-VIs.

Figure 4: Reconstruction of MNIST handwritten digits. Left column shows the original digits. Right column shows the reconstructed digits. (a) is from IW-ELBO loss. (b) is from IW-CUBO ($n = 2$) loss. (c) is from IW-RVB ($\alpha = 3$) loss. (d) is from IW-TVB loss. (e) is from custom $f_{c1}$-variational bound loss, and (f) is from custom $f_{c2}$-variational bound loss.

Figure 5: Generation of MNIST handwritten digits. (a) is from IW-ELBO loss. (b) is from IW-CUBO ($n = 2$) loss. (c) is from IW-RVB ($\alpha = 3$). (d) is from IW-TVB loss. (e) is from custom $f_{c1}$-variational bound loss, and (f) is from custom $f_{c2}$-variational bound loss.

Figure 6: Generation of Caltech 101 silhouettes. (a) is from IW-ELBO loss. (b) is from IW-CUBO ($n = 2$) loss. (c) is from IW-RVB ($\alpha = 3$). (d) is from IW-TVB loss. (e) is from custom $f_{c1}$-variational bound loss, and (f) is from custom $f_{c2}$-variational bound loss.

Figure 7: Generation of Frey Face. (a) is from IW-ELBO loss. (b) is from IW-CUBO ($n = 2$) loss. (c) is from IW-RVB ($\alpha = 3$). (d) is from IW-TVB loss. (e) is from custom $f_{c1}$-variational bound loss, and (f) is from custom $f_{c2}$-variational bound loss.

Figure 8: Generation of Omniglot alphabets. (a) is from IW-ELBO loss. (b) is from IW-CUBO ($n = 2$) loss. (c) is from IW-RVB ($\alpha = 3$). (d) is from IW-TVB loss. (e) is from custom $f_{c1}$-variational bound loss, and (f) is from custom $f_{c2}$-variational bound loss.

## Footnotes

[5]When deriving the ELBO in (19), we also multiplied the $f$-variational bound by $-1$.

[6](full name, #instances, #attributes) of six new benchmarks are provided: *Airfoil* (Airfoil Self-Noise, 1503, 6), *Aquatic* (QSAR Aquatic Toxicity, 546, 9), *Building* (Residential Building Data Set, 372, 105), *Fish Toxicity* (QSAR Fish Toxicity, 908, 7), *Real Estate* (Real Estate Valuation Data Set, 414, 7), and *Stock* (Stock Portfolio Performance, 315, 12).