[Reviews · NeurIPS 2020]

Review 1

Summary and Contributions: The authors derive VI for general-form f-divergences unifying numerous previous VI approaches. They introduce surrogate f-divergence along with a set of mathematical objects necessary for its characterization and using the surrogate f-divergence they derive a VI objective and sandwich bounds for evidence. Finally, they describe stochastic optimization, importance weighting, and Monte Carlo gradients for the generalized VI framework. The empirical evaluation includes examples of BNNs and VAEs.

Strengths: The publication provides a complete description and characterization of a general framework for VI that unifies several recent approaches. It extends the understanding of VI and the unified formalization may significantly help in future developments of the method, for example, by enabling constructing new bounds.

Weaknesses: The publication is complex and the authors do not provide much intuition regarding mathematical results that could help to broaden its potential impact. Furthermore, their approach establishes bases for potential future research but adds relatively little value for today's practitioners, for example, no quantitative improvements were really achieved in the experiments. Minor: Although useful for completeness, some parts of Section 3 (mostly 3.3 and the first half of 3.2) repeat previous results.

Correctness: The theoretical claims seem to be correct, although, I checked only part of the proofs provided in Supplementary Materials. The empirical methodology is correct.

Clarity: The paper is well written. It has a good organizational structure and clear flow: from basic definitions and properties, through the formulations for VI to experimental evaluations.

Relation to Prior Work: The authors provide a broad discussion of VI and previous methods over which f-divergence VI generalizes, but they do not go into detail. The publication lacks a deeper discussion of [15,17,18], making it hard to understand what are the previous results regarding f-divergences and VI.

Reproducibility: Yes

Additional Feedback:


Review 2

Summary and Contributions: This paper proposes a general variational inference (VI) framework based on f-divergence, termed f-VI. Three main contributions are delivered, including (1) a general VI framework (f-VI) that contains many existing VI methods as special cases; (2) a sandwich estimate of the evidence based on f-VI, with importance-weighted techniques also revealed to further tighten the estimate; (3) a mean-field f-VI (similar to the classic mean-field VI) is also presented. The paper is well-written. However, the experiment section is very weak.

Strengths: The presented techniques are sound. The theoretical contributions seem to be novel and significant.

Weaknesses: The advantages of the proposed techniques over existing ones are not clear. Although the proposed f-VI is indeed general, why and how to use it is not addressed. The experiments are too weak.

Correctness: The presented techniques are correct.

Clarity: The paper is well-written.

Relation to Prior Work: Although the proposed f-VI contains many existing methods as special cases, the corresponding advantages are not clearly discussed.

Reproducibility: Yes

Additional Feedback: Based on Eq. (2), to derive Eq. (8) is straightforward, as revealed in Line 405 of the SM. Why spent so much effort to circle around from a dual space? Eq. (7) and Line 131, a typo exists in the subscripts. The equation under Line 143 is not right. Line 179, why treating the parameters phi as latent variables z helps reduce variance? Line 200. The word “two-dimensional” is confusing, because in general, noise samples need not be 2-dimensional. One big question, why and how to use the proposed f-VI? In the experiments, it seems the commonly used KL-div-based VI works the best in general. So, what are the advantages of f-VI over existing methods, in addition to being general? ==== After Rebuttal ==== I have read all the comments and the author feedback carefully. Despite the clear novelty/contributions of this manuscript, I decide to decrease my score from a 6 to a 5. The main reasons include: (1) why and how to use the proposed f-VI in practice remain unclear after the rebuttal, leaving this manuscript incomplete to some extent; it's a pity that the authors didn't even try to provide intuitive solutions. (2) the current experimental results didn't convince me that the f-VI is indeed useful in practice, while the newly added experiments are not accessible. (3) despite I like the paper a lot, the authors didn't do a successful job in addressing my concerns in the rebuttal.


Review 3

Summary and Contributions: The paper introduces a general f-divergence variational inference framework with a theoretical analysis. Moreover, the paper extends their f-divergence framewrk to current variational inference methods. Simple experiments are provided to demonstrate the idea.

Strengths: The paper is well-written and presents a detailed description of the analysis and proofs (the supplemental material is pretty detailed and well-written as well). The idea of f-divergence is not new, but the incorporation of a general framework (with the proofs) is an interesting contribution to the community.

Weaknesses: The empirical demonstration of the paper is rather weak. The authors demonstrate their method on a toy dataset and on the UCI ML repository. Additional results on more challenging datasets would be interesting. The dataset will depend on the task to use or validate the method on, so it is not the place to extensively cite the datasets here.

Correctness: The proofs seem correct. I did not find a contradiction or error on them.

Clarity: The paper is well written and easy to follow. The proofs and descriptions on the supplementary material are really helpful.

Relation to Prior Work: The related work is discussed superficially. However, due to space, I'm not sure where they could extend on it. Perhaps on the supplementary material.

Reproducibility: Yes

Additional Feedback: ### Comments after rebuttal While I miss the experimental results to support and improve the paper, the authors did provide a solid theoretical paper. I don't think that this work is a rejection, but also do not consider it a top paper due to the lack of more thorough experiments. Overall a good contribution for the conference. ###


Review 4

Summary and Contributions: The authors propose a novel variational inference framework based on the f-divergence. To this end they developed a surrogate divergence based on the f-divergence. This surrogate divergence allows for a general f-variational upper bound on the dual function f* of the evidence which allows to recover existing variational bounds as special cases. Composition of the f-variational bound with the inverse of the dual function f* results in an evidence upper or lower bound for monotonically increasing or decreasing intervals of f* respectively. As the f-variational bounds generally depends non-trivially on the evidence the authors propose to approximate the likelihood term using mini-batches and refer to the resulting algorithm as f-variational inference. For more tractable computation they derive update rules for fully factorized proposals. The authors evaluate f-VI in three experiments: 1 - A synthetic data set where they recover existing VI approaches by optimizing the respective bounds 2 - Bayesian linear regression on UCI MLR where they compare a custom f-VI instance and known VI instances based on the RMSE. 3 - A VAE trained on MNIST where instances are compared based on cross entropy and visually based on their reconstructions.

Strengths: The idea of a generalized f-variational bound which recovers existing approaches is appealing and offers a framework to theoretically and empirically evaluate novel and existing f-functions. The theoretical claims seem to be sound and are supported by detailed supplementary material.

Weaknesses: The ability to evaluate custom f-functions easily in one common framework is one of the strong points of f-VI. However, as f-VI recovers existing VI methods it is unclear to what extend the experiments show the "effectiveness of f-VI". It is already established that commonly used f-functions perform well. I understand that custom f-functions should be evaluated against the well established f-function but I'm surprised that the second and third experiment only test a single custom f-function. An experiment aiming to empirically find suitable functions f and g which induce tighter bounds on the evidence (using corollary 2 as described in the paper) seems to be a better suited experiment to demonstrate the usefulness of the proposed framework.

Correctness: To the best of my knowledge the author's claims and empirical methodology are correct. (I have verified the main claims to the best of my capabilities but have not worked though all of the supplementary material.)

Clarity: While the paper is overall well organized and the technical part of the paper can be read and understood without difficulties, the writing can certainly be improved and the paper would benefit from additional proof-reading for typos, English style, and grammar. The introduction lacks clarity and does not elaborate on some important points. This is especially important as the paper does not have a dedicated related work section or background on VI. I understand that the paper assumes some familiarity with VI but still think it could be written clearer and be more accessible without making it too lengthy. Some points I noticed: l.20 - It is mentioned that MCMC methods are not suited for modern machine learning problems without any quantification l.20, l.28 - Terms like MCMC, traditional VI, or scalable VI, are used without any further explanation l.28 - What are the improvements over 'traditional' VI and MCMC methods? Briefly stating or at least naming them would enhance the point that the introduction is trying to make. Some typos and style/grammar related issues I noticed on the first page: l.03 - crafty l.09 - reparameterzation -> reparameterization l.10 - the importance-weighted method -> importance weighting l.18 - has been -> have been l.22 - VI becomes a good alternative to Bayesian; VI is not an alternative to Bayesian inference but can used to performing Bayesian inference l.23 - family of approximate (or recognition) densities; I suppose Q is a family of densities not a family of approximations to densities l.28 - Recent advances in VI showed improvements over traditional VI [4, 5], scalable VI [6–8], and tighter variational bounds -> such as tighter variational bounds? l.38 - the f-divergence

Relation to Prior Work: The authors clearly state their contributions. Important prior work is pointed out and cited but the discussion could be more exhaustive.

Reproducibility: Yes

Additional Feedback: **Update** I've carefully read through the author's rebuttal and the other reviews. Including two additional custom f-functions in their VAE and BNN experiments and discussing strategies for generating custom f-functions will likely make the experiment section stronger but to what extend is unclear as the authors did not include any results in the rebuttal. That said, the paper makes a solid theoretical contribution that is of interest to the community. Hence, I increase my score to a 7.

[Author Response · NeurIPS 2020]

We thank the reviewers for their constructive feedback and unanimous acceptance of our theory on general $f$-VI.
We notice that *a big concern among all reviewers is about our experiments, as the reviewers commented that the*
*experiments i) did not interpret the improvements of $f$-VI against the well-established VI methods, ii) need to be tested*
*on more challenging datasets, and iii) should include more custom $f$-functions.* We would like to mention that our main
contribution is to provide a solid theoretical foundation and two (stochastic and mean-field) optimization schemes for
VI subject to all $f$-divergences, and our experiments then serve to verify the correctness and feasibility of these results.
While proposing a specific $f$-divergence VI that surpasses the existing VI methods is a significant task, the workload of
it deserves an independent and thorough study as in [16, 17], and the current paper can guide and facilitate this task by
offering a general $f$-VI framework. Meanwhile, our experiments (Section 4.2 and 4.3) suggest that the performance
of different $f$-divergence VIs varies by the training model as well as the dataset; *e.g.* the custom $f$-divergence VI
showed some quantitative improvements against other well-established $f$-divergence VIs in the BNN experiment on
Fish Toxicity and Stock datasets, while it slightly underperformed in the other empirical examples; similar results were
also reported in [2, 3], which compared the KL-VI and some Rényi's $\alpha$-VIs. Hence, a (custom) $f$-divergence VI that
consistently dominates other well-studied $f$-VIs is almost unforeseeable. However, as per your request, we will add the
following new examples:
1) Two new custom $f$-functions, $f^*(t) = |t - 1|$ for total variation distance and $f^*(t) = -\log(2t + 1) - \log t + \log 3$,
are tested in the BNN experiment of Section 4.2 and the VAE experiment of Section 4.3. The results are appended in
the Supplementary Material (SM). Some strategies for generating a valid custom $f$-function are also supplemented.
2) Three additional datasets, Frey Face, Caltech 101 Silhouettes and Omniglot, are tested in the VAE experiment of
Section 4.3. Reconstruction errors and some reconstructed and generated images are added and compared.

*A minor comment from all reviewers is about the introduction section, as the reviewers pointed out that the introduction*
*i) did not have a dedicated related work section or background on VI, and ii) lacked a deeper discussion when reviewing*
*prior work on VI and $f$-VI, such as* [15, 17, 18]. We referred the readers without any experience in VI to two recent
survey papers [11, 12] for VI background. Prior work on $f$-VI either investigated only a specific family of $f$-divergence
[16, 17] or circumvented the fundamental setting of $f$-VI [18], *i.e.* minimizing $D_f(q(z)|p(z))$. Hence, none of them
can unify the existing VI methods and also be applicable to all $f$-divergences. More deeper discussions on the prior
work and VI background will be added in either the introduction or the SM. *Other comments with regard to the Boarder*
*Impact and writing* will be addressed in the camera-ready paper. The following is our response to the individual
questions raised by the reviewers:

**R2:** *Although useful for completeness, some parts of Section 3 (mostly 3.3 and the first half of 3.2) repeat previous*
*results.* - The general $f$-VI algorithms presented in Section 3 are new and have not been reported before, although with
proper $f$- or $f^*$-functions assigned, they restore many well-known Bayesian approximation algorithms, *e.g.* examples
in Section C and D. New Bayesian approximation or VI algorithms can also be generated from the results in Section 3,
if we assign the $f$-VI algorithms in Section 3 with new (custom) $f$- or $f^*$-functions.

**R2:** *What is the intuition regarding mathematical results that could help to broaden the potential impact of $f$-VI?*
**R3:** *Why and how to use the proposed $f$-VI? In the experiments, it seems the commonly used KL-VI works the best in*
*general. So, what are the advantages of $f$-VI over existing methods, in addition to being general?* - We will answer the
preceding two comments collectively. An explicit advantage (or impact) of $f$-VI is that it allows to perform Bayesian
approximation or VI with more variety of divergences, which could potentially bring us sharper variational bounds,
faster convergence rates, smaller RMSE and reconstruction error as in our experiments. The empirical performance
of different $f$-divergence VIs varies by the training model and datasets, while the logarithmic KL-function makes the
KL-VI more numerically stable and accurate when $p(z, \mathcal{D})/q(z)$ is tiny, which can be a guideline for picking $f$ or
$f^*$-functions.

**R3:** *Based on Eq. (2), to derive Eq. (8) is straightforward, as revealed in Line 405 of the SM. Why spent so much effort*
*to circle around from a dual space?* - Eq. (8) is derived by minimizing the reverse $f$-divergence, while Eq. (18) is from
the forward $f$-divergence. Eqs. (8) and (18) are different, once you expand the dual functions in Eq. (8) and consider
the constraints on $f$- and $f^*$-functions. Since the existing VIs based on the reverse divergences generally have better
statistical properties [18, 20], we derived the $f$-variational bound Eq. (8) in a dual space, which also makes our results
more compatible and consistent with the existing VI algorithms. However, both Eq. (8) and (18) are essential parts of
our $f$-VI framework, as the mean-field update rules Eq. (15) and (16) are respectively derived from Eqs. (8) and (18).

**R3:** *Line 179, why treating the parameters $\phi$ as latent variables $z$ helps reduce variance?* - Treating the parameter $\phi$
in $p_\phi(z, \mathcal{D})$ as latent variables does not necessarily help reduce the variance. We will withdraw this statement in the
camera-ready paper.

**R3:** *Line 200. The word "two-dimensional" is confusing, because in general, noise samples need not be 2-dimensional.*
- "Two-dimensional" describes the size of noise samples $\{\varepsilon_{k,1:L}\}_{k=1}^K$ in the importance-weighted estimators, Eq. (14)
and Line 143. We will withdraw this word in the camera-ready paper.

[Meta-Review · NeurIPS 2020]

The reviewer agreed that this is a solid theoretical contribution, although the experimental part could be improved.